# Apolipoprotein E intersects with amyloid-$\beta$ within neurons

Sabine C Konings[1,2] , Emma Nyberg[1] , Isak Martinsson[1], Laura Torres-Garcia[1] , Oxana Klementieva[2], Claudia Guimas Almeida[3] , Gunnar K Gouras[1]

Apolipoprotein E4 (ApoE4) is the most important genetic risk factor for Alzheimer's disease (AD). Among the earliest changes in AD is endosomal enlargement in neurons, which was reported as enhanced in ApoE4 carriers. ApoE is thought to be internalized into endosomes of neurons, whereas $\beta$-amyloid (A$\beta$) accumulates within neuronal endosomes early in AD. However, it remains unknown whether ApoE and A$\beta$ intersect intracellularly. We show that internalized astrocytic ApoE localizes mostly to lysosomes in neuroblastoma cells and astrocytes, whereas in neurons, it preferentially localizes to endosomes–autophagosomes of neurites. In AD transgenic neurons, astrocyte-derived ApoE intersects intracellularly with amyloid precursor protein/A$\beta$. Moreover, ApoE4 increases the levels of endogenous and internalized A$\beta_{42}$ in neurons. Taken together, we demonstrate differential localization of ApoE in neurons, astrocytes, and neuron-like cells, and show that internalized ApoE intersects with amyloid precursor protein/A$\beta$ in neurons, which may be of considerable relevance to AD.

## Introduction

The apolipoprotein E (ApoE) allele exists as three major isoforms in humans: ApoE2, ApoE3, and ApoE4, with ApoE3 being the most common form, followed by ApoE4. ApoE4 is the most important genetic risk factor for Alzheimer's disease (AD) [1, 2]. In contrast, ApoE2 protects against AD. ApoE in the brain is made predominantly by astrocytes, but is also generated by microglia and under conditions of stress by neurons [3, 4]. Although the most critical mechanism(s) by which ApoE4 raises the risk of AD remains to be determined, various hypotheses have been proposed. A tight correlation of ApoE4 with amyloid pathology is well known, but the mechanism(s) whereby ApoE4 impacts amyloid pathology in AD remains unclear. Previous research suggests a role of ApoE4 in the early stages of AD, before amyloid plaques [5, 6]. The presence of ApoE4 increases the number of dystrophic neurites around plaques and affects initial plaque density, but does not appear to influence plaque load after initial plaque formation [5, 6], suggesting a role of ApoE4 in the early cellular phase of AD before amyloid plaques.

Endosomal enlargement in neurons is one of the earliest changes associated with AD [7, 8]. Postmortem brains from late-onset AD patients showed significantly enlarged neuronal endosomes compared with age-matched control brains [7], a finding that has also been described in vitro in IPSC-induced neurons from AD patients [9] and as induced by AD genetic risk factors [10, 11]. The mean endosomal size was reported as even more increased in postmortem AD human brains from ApoE4 carriers compared with non-ApoE4 carriers, suggesting an enhancing effect of ApoE4 on endosomal enlargement [7]. The exact mechanism(s) underlying endosome dysfunction in AD, in particular in relation to ApoE4, remains poorly understood. Experimental studies have reported endosomal dysregulation induced by ApoE4 and ApoE4-induced impairment in endosome recycling [12, 13, 14, 15]. Added recombinant ApoE was described to localize in neurons to the endosomal system at Rab5-positive early endosomes and cathepsin D-positive late endosomes/lysosomes [16]. However, studies on the endosomal localization of more physiological lipidated ApoE isoforms in neurons are lacking.

Before plaque formation, intraneuronal A$\beta$ accumulates in endosomes [17, 18, 19], in particular, in late endosomal multivesicular bodies near synaptic compartments [18]. In addition, BACE1-induced APP cleavage is favored under acidic pH conditions as present in endosomes [20], APP trafficking, and A$\beta$ production occur in endosomes [11, 21, 22]. As A$\beta$ is generated [21, 23] and initially aggregates [18] in acidic endosomes, ApoE4 might facilitate the optimal conditions for A$\beta$ production by blocking endosome recycling or provide a more favorable lipid environment for A$\beta$ aggregation. ApoE was reported to trigger APP endocytosis and subsequent A$\beta$ production after binding to ApoER2 receptors, with ApoE4 increasing A$\beta$ production more than ApoE2 and ApoE3 [24]. In vivo data also support an association of ApoE4 with increased

---

[1]Experimental Dementia Research Unit, Department of Experimental Medical Science, Lund University, Lund, Sweden   [2]Medical Microspectroscopy, Department of Experimental Medical Science, Lund University, Lund, Sweden   [3]iNOVA4Health, NOVA Medical School | Faculdade de Ciências Médicas, Universidade Nova de Lisboa, Lisboa, Portugal

Correspondence: gunnar.gouras@med.lu.se

neuron levels of Aβ, as human ApoE target replacement mice transduced with lentiviral Aβ$_{1-42}$ showed increased intracellular Aβ deposition in ApoE4 compared with ApoE3 mouse brains (25). On the other hand, in another study, mouse brain knock-in with the different human ApoE isoforms did not reveal alterations in APP levels or processing (26), whereas crosses of AD transgenic with ApoE mice showed elevated intraneuronal Aβ with ApoE4 (27).

It is possible that internalized ApoE and APP and/or Aβ intersect and even interact within endosomes. ApoE localizes with high-molecular weight Aβ oligomers in the TBS-soluble fraction of human AD brain (28), and ApoE co-deposits with amyloid plaques (29), suggesting that ApoE and Aβ also co-localize at later stages of AD, but whether ApoE and APP/Aβ are intersecting at an intracellular level remains less clear. ApoE and antibody 4G8-positive Aβ/APP were reported to be present in the same cytoplasmic granules in postmortem AD brains (30), although the antibody used did not distinguish Aβ from its abundant precursor APP. A subsequent study using Aβ$_{42}$ C-terminal-specific antibodies noted localization of ApoE to AD-vulnerable neurons with marked intraneuronal Aβ$_{42}$ accumulation in human brains with early AD pathology (31). More recently, ApoE and Aβ were reported to be present in the same synapses of human AD brains (32, 33).

Despite ApoE4 being established as the most important genetic risk factor for AD and neurons selectively degenerating in the disease, the subcellular localization of ApoE and its potential relation to Aβ/APP biology in neurons has not been explored. The aim of this study was to examine the subcellular localization of ApoE3 and ApoE4. In addition, the localization of the ApoE isoforms with intracellular Aβ/APP was examined. We demonstrate that internalized ApoE predominantly traffics to lysosomes in primary astrocytes and N2a neuroblastoma cells, but not in primary neurons where it localizes more to endosomes and autophagosomes of neurites. Furthermore, ApoE co-localized with Aβ/APP β-C-terminal fragments (APP-βCTFs), with ApoE4 treated-neurons showing more prominent intraneuronal Aβ.

## Results

### ApoE is internalized into the endosome–lysosome system of N2a cells

The endosome–lysosome system is considered to be one of the earliest cellular sites affected in AD (7, 34, 35). ApoE4 was previously shown to impact neuronal endosomes, including endosomal recycling (12) and endosomal morphology (14); however, the subcellular biology of ApoE remains poorly studied. In order to better define the subcellular localization of ApoE, N2a neuroblastoma cells were initially treated with physiological levels (2.5 μg/ml) of recombinant human ApoE3 or ApoE4 for 15 min and 4 h (Fig 1A). After 4 h more, N2a cells internalized recombinant ApoE4 compared with ApoE3 (Figs 1B and S1A). Specifically, ApoE3 was taken up by 23.0% and ApoE4 by 33.3% of N2a cells (Fig 1C, P = 0.024). Moreover, ApoE4 treatment increased the median number of ApoE puncta per N2a cells by 33.4% compared with ApoE3 treatment (Fig 1D, P = 0.034). The area of ApoE puncta in ApoE4-treated N2a cells showed an

increasing trend, although this difference was not significant (Fig 1E, P = 0.0767). The puncta of internalized recombinant ApoE showed a vesicle-like pattern (Fig 1F), consistent with the endosomal–lysosomal system, although interestingly, most of the ApoE labeling appeared polarized and near to the Golgi apparatus and microtubule-organizing center. Due to endogenous mouse ApoE expression by N2a cells and the fact that the ApoE antibody 16H22L18 to some extent also detects mouse ApoE, we were unable to distinguish differences in ApoE in vehicle-treated N2a cells from human ApoE-treated cells at the shorter (15 min) time point (data not shown). For this reason, further characterization of human ApoE in neuronal cells was performed using the 4 h time point, at which time, the internalized ApoE was clearly above the lower endogenous mouse ApoE labeling.

To better define the presumed endosomal–lysosomal trafficking of internalized recombinant ApoE in N2a cells, we co-labeled ApoE-treated N2a cells for ApoE and subcellular markers. Due to the vesicle-like localization of ApoE, we started with analyzing ApoE co-localization with antibodies to LAMP1, a lysosomal marker, which also detects late endosomes and autophagosomes, and to Rab7, a late endosomal marker, which also can be detected in autophagosomes. At 4 h, 36% of internalized human ApoE3 and 30% of ApoE4 co-localized with LAMP1 (overlap of ApoE-positive pixels with LAMP1-positive pixels; Figs 1G and I and S1B), suggesting lysosome, late endosome, and/or autophagosome localization of internalized recombinant ApoE3 and ApoE4 in N2a cells. However, there was no statistically significant difference in ApoE and LAMP1 co-localization between ApoE3 and ApoE4 (Figs 1G and I, P = 0.2744). In comparison with LAMP1, a lower proportion of human ApoE puncta co-localized with Rab7 in N2a cells after 4 h, although there was no significant difference between ApoE3 compared with ApoE4 (Figs 1H and J; P = 0.3392). We note that Rab7 labeling distributed more widely throughout the cell, whereas most ApoE and LAMP1 localized in the perinuclear region, likely near to the Golgi/microtubule organizing center. Overall, the LAMP1-positive, mostly Rab7-negative, co-labeling of internalized human ApoE suggests that human ApoE after 4 h has mostly trafficked beyond late endosomes to lysosomes in N2a cells. To assess whether internalized ApoE might also be transported to the Golgi apparatus, the presence of ApoE at the cis- and trans-Golgi apparatus was assessed. Despite preferential labeling of internalized ApoE3 and ApoE4 near to GM130-labeled cis-Golgi apparatus (Fig S2A), no clear direct co-localization was seen of human ApoE with the antibody against GM130 or with the trans-Golgi marker TGN38 (Fig S2). Altogether, internalized recombinant ApoE3 and ApoE4 in N2a cells are most notably localized to lysosomes after 4 h incubation.

Although recombinant ApoE is generally considered to be poorly lipidated, previous research suggests that cell media containing FBS can provide ApoE with lipids (36), and we note that our N2a cell media contain FBS. Because ApoE is the main lipid and cholesterol carrier in the brain, we next examined whether cholesterol localizes to similar subcellular compartments as ApoE. Interestingly, filipin, a dye staining free cholesterol, revealed that in N2a cells cholesterol, analogous to recombinant ApoE, localized to LAMP1-positive vesicles (Fig 1K).

Cellular trafficking of ApoE might however differ depending on the lipidation of ApoE. Therefore, we next studied the cellular

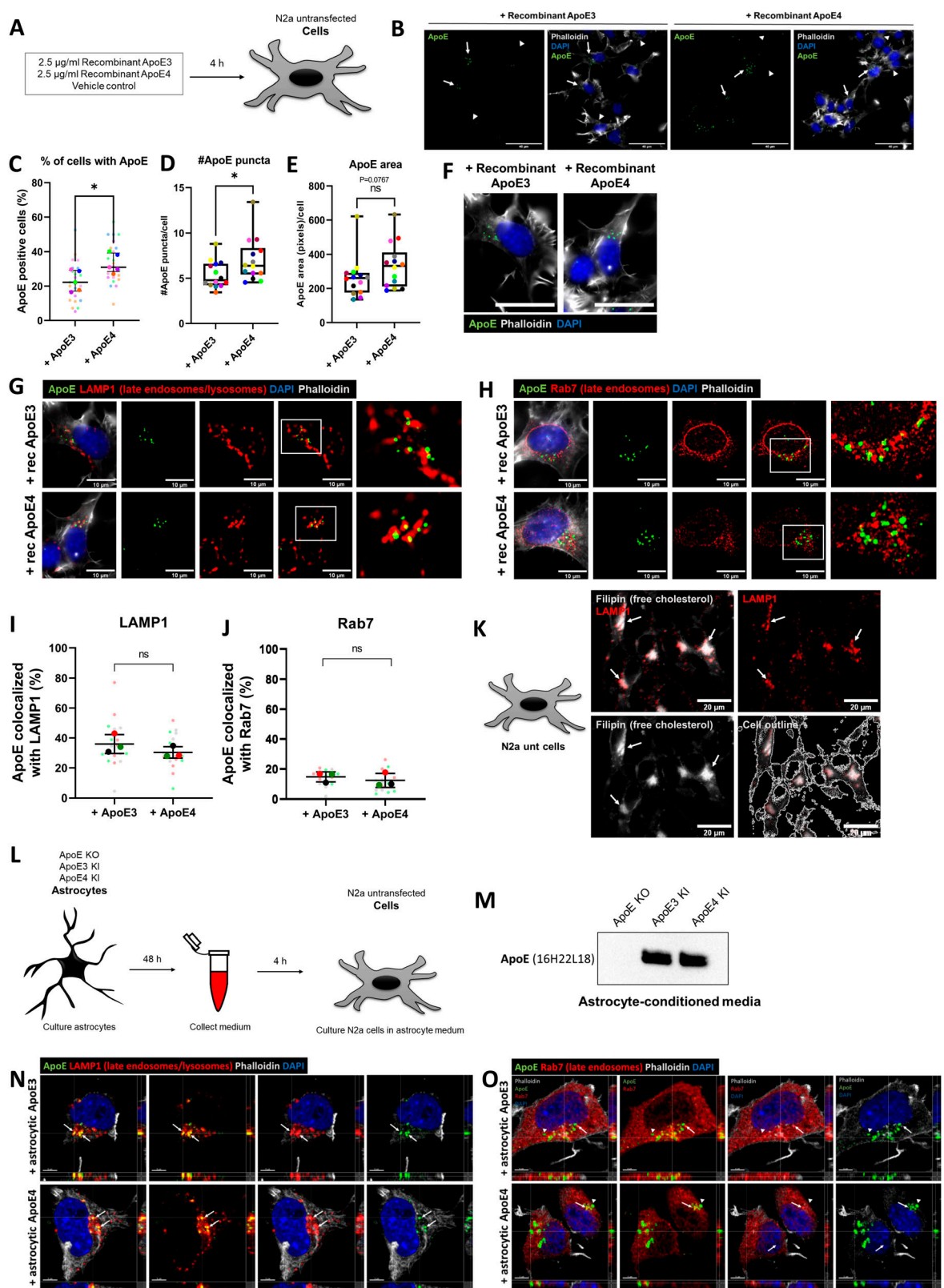

**Figure 1. Recombinant and astrocyte-derived ApoE are internalized into the endosome–lysosome system in N2a cells.**
**(A)** Schematic overview of the 4 h treatment with 2.5 μg/ml recombinant ApoE3, ApoE4 or vehicle control in N2a cells. **(B)** Representative epifluorescence images of N2a cells treated with recombinant ApoE3 or ApoE4 for 4 h showing an overview of human ApoE internalization in N2a cells. The N2a cells were labeled for ApoE (green), DAPI (blue), and phalloidin (grey). Cells showing internalized human ApoE are indicated by arrows, and cells negative for ApoE are indicated by arrowheads. Scale bar is 40 μm.

localization of ApoE under more physiological conditions by using ApoE particles obtained from humanized ApoE3-knock-in or ApoE4-KI primary astrocytes (37). Note that these human ApoE-targeted replacement mice no longer express mouse ApoE. Astrocytes are the main source of ApoE in the brain (38) and astrocyte-derived ApoE is lipidated (39) and transported to other cell types including neurons. Human ApoE3 and ApoE4 astrocyte-conditioned media were collected from ApoE3- and ApoE4-KI mouse primary astrocytes and subsequently used to treat N2a cells for 4 h (Fig 1L). Human ApoE was readily detected by Western blot in media collected from ApoE3- and ApoE4-KI mouse astrocytes, but, as expected, not from ApoE KO astrocytes (Fig 1M), confirming ApoE3 and ApoE4 conditioned media as a useful source for human ApoE. Like recombinant human ApoE, added astrocyte-derived human ApoE3 and ApoE4 after 4 h co-localized substantially with LAMP1-positive vesicles in N2a cells (Figs 1N and S1C), and showed lower levels of co-localization in the more numerous Rab7-positive late endosomes (Fig 1O). These findings confirm that internalized human ApoE3 and ApoE4, recombinant or derived from primary astrocytes, mostly traffics to lysosomes of N2a cells.

### Internalized ApoE in primary neurons preferentially localizes to endosomes and autophagosomes in neurites

Even though N2a cells are considered to be neuron-like cells, their morphology and cellular physiology are quite different from mature neurons. To study the localization of astrocyte-derived human ApoE in mature primary neurons, the subcellular localization of human ApoE was examined after adding conditioned astrocyte ApoE3 or ApoE4 media to ApoE KO primary brain cultures for 4 h (Fig 2A). ApoE KO brain cultures were used to allow only the visualization of the subcellular localization of added astrocyte-derived human ApoE without the presence (and background signal) of endogenous mouse ApoE. Because in N2a cells both recombinant and astrocyte-derived ApoE localized mostly to LAMP1-positive vesicles (Fig 1), we initially studied neuronal cell bodies, where lysosomes are predominantly located in neurons (40, 41, 42). Surprisingly, after 4 h of treatment internalized astrocyte-derived ApoE3 and ApoE4

localized only to a limited extent to neuronal cell bodies where, unlike N2a cells, ApoE was not found to co-localize with LAMP1-positive vesicles (Fig 2B; ApoE KO media control in Fig S1D), suggesting limited trafficking of added astrocytic ApoE to lysosomes in neuron soma. Internalized human ApoE puncta were also not seen to overlap with Rab7 in neuronal soma (Fig 2C). Of note, most ApoE labeling near cell bodies actually appeared to be in the periphery and/or just outside the neuron cell soma.

Remarkably, in primary neurons (DIV 19), most of the added astrocyte-derived ApoE after 4 h was seen in neurites rather than neuronal cell bodies (Fig 2D–G). In contrast to neuronal soma, added astrocytic ApoE3 and ApoE4 after 4 h were seen to be present in LAMP1-positive vesicles in neurites, which in our neurons should represent late endosomes and/or amphisomes (Fig 2D; ApoE KO media control in Fig S1E) (43, 44). In line with this, the internalized human ApoE also partially co-localized with Rab7 in neurites (Fig 2E). However, the majority of added astrocytic ApoE did not co-localize with either LAMP1- or Rab7-positive puncta, highlighting other cellular site(s) of ApoE in neurites.

Since internalized astrocytic ApoE3 and ApoE4 labeled in a vesicular pattern, the subcellular localization of ApoE in neurites was further studied using additional markers, with EEA1 to identify early endosomes, and LC3β to label autophagosomes. After 4 h, the added astrocytic ApoE was detected in both LC3β-positive autophagosomes (Fig 2F) and EEA1-positive early endosomes (Fig 2G) at comparable levels with LAMP1- and Rab7-positive vesicles (Fig 2D and E). Altogether, these findings suggest that astrocyte-derived ApoE, after internalization into primary neurons, is present in the endosome-autophagy system of neurites.

Since levels of internalized ApoE in lysosomes of neuron soma were not as apparent as in N2a cells, we wondered whether this might also be caused by degradation of astrocyte-derived ApoE by neuronal lysosomes. To assess possible degradation of ApoE by lysosomes in neurons, we inhibited lysosomal degradation using 10 nM bafilomycin A1 (BafA1) starting 1 h before addition for 4 h of ApoE3 or ApoE4-conditioned media. BafA1 inhibits lysosomal function by blocking lysosomal acidification via v-ATPase (45, 46) (Fig 2H). The size of LAMP1-positive vesicles in neurites increased

---

(C) Quantification of the percentage of ApoE-positive N2a cells within the entire culture. Recombinant ApoE4-treated N2a cells showed a significantly higher percentage of ApoE-positive cells (10.3% increase for ApoE4; t test, P = 0.024; number of cultures per treatment group: 5 [big data points in graph]; within each culture, 4–5 regions were analyzed [small data points in graph]). (D, E) Quantification of the number of ApoE puncta (D) and puncta area (E) per N2a cell. ApoE4 treatment significantly increased the median ApoE puncta number by 33.4% (mean puncta increase: 1.6) (Mann–Whitney test, P = 0.034) and showed a trend in the ApoE puncta area increase (by 27.5%) for ApoE4 compared with ApoE3 treatment (Mann–Whitney test, P = 0.0767; number of cultures per treatment group: 14, number of images taken within each culture: 1–7). (F) Higher magnification image of ApoE3- and ApoE4-treated N2a cells from Fig 1B. Internalized recombinant ApoE3 and ApoE4 puncta detected in N2a cells show a vesicle-like pattern. Scale bar represents 40 μm. (G) Representative images obtained by epifluorescence microscopy of N2a cells incubated with recombinant ApoE3 or ApoE4 for 4 h. The cells were labeled for ApoE (green), late endosomal/lysosomal marker LAMP1 (red), DAPI (blue), and phalloidin (grey). The scale bar represents 10 μm. (H) Representative epifluorescence images of recombinant ApoE3- and ApoE4-treated N2a cells labeled for ApoE (green), late endosomal marker Rab7 (red), DAPI (blue), and phalloidin (grey). Scale bar is equal to 10 μm. (I) Quantification of Fig 1D, showing the co-localization levels (in percentages) of ApoE with LAMP1. After ApoE3 and ApoE4 treatment, on average 35.8% and 30.7% of ApoE, respectively, co-localized with LAMP1 (t test on difference between ApoE3- and ApoE4-treated N2a cells, P = 0.2744; number of cultures [big data points in graph]: 3; number of cells analyzed per culture [small data points in graph]: 5). (J) Quantification of ApoE-Rab7 co-localization of Fig 1F. 14.2% and 12.1% of the ApoE co-localized with Rab7 in N2a cells after ApoE3 and ApoE4 treatment, respectively (t test on difference between ApoE3- and ApoE4-treated N2a cells, P = 0.3392; number of cultures [big data points in graph]: 3; number of cells analyzed per culture [small data points in graph]: 5). (K) Representative images taken by epifluorescence microscopy of N2a cells showing LAMP1 (red) overlaps with Filipin (grey), a free cholesterol dye. Scale bar is 20 μm. (L) Schematic representation of astrocyte-conditioned media collection from ApoE KO, ApoE3 KI, and ApoE4 KI primary astrocytes, followed by media incubation of N2a cells for 4 h. (M) Representative Western blot of secreted human ApoE proteins detected in astrocyte-conditioned media after culturing 48 h with ApoE KO, ApoE3 KI, and ApoE4 KI primary astrocytes. (N, O) Representative orthogonal images obtained by confocal microscopy showing ApoE (green) co-localizing with LAMP1 (red) (N) and to a lower extent also with Rab7 (red) (O) in ApoE3 and ApoE4 astrocyte-conditioned media–treated N2a cells. The cells were further labeled for DAPI (blue) and phalloidin (grey). (N, O) Arrows indicate ApoE puncta inside LAMP1- (N) and Rab7- (O) positive vesicles. Scale bars represent 5 μm. Data are expressed as mean ± SD. ns, nonsignificant, Rec ApoE, recombinant ApoE. See also Fig S1A–C.

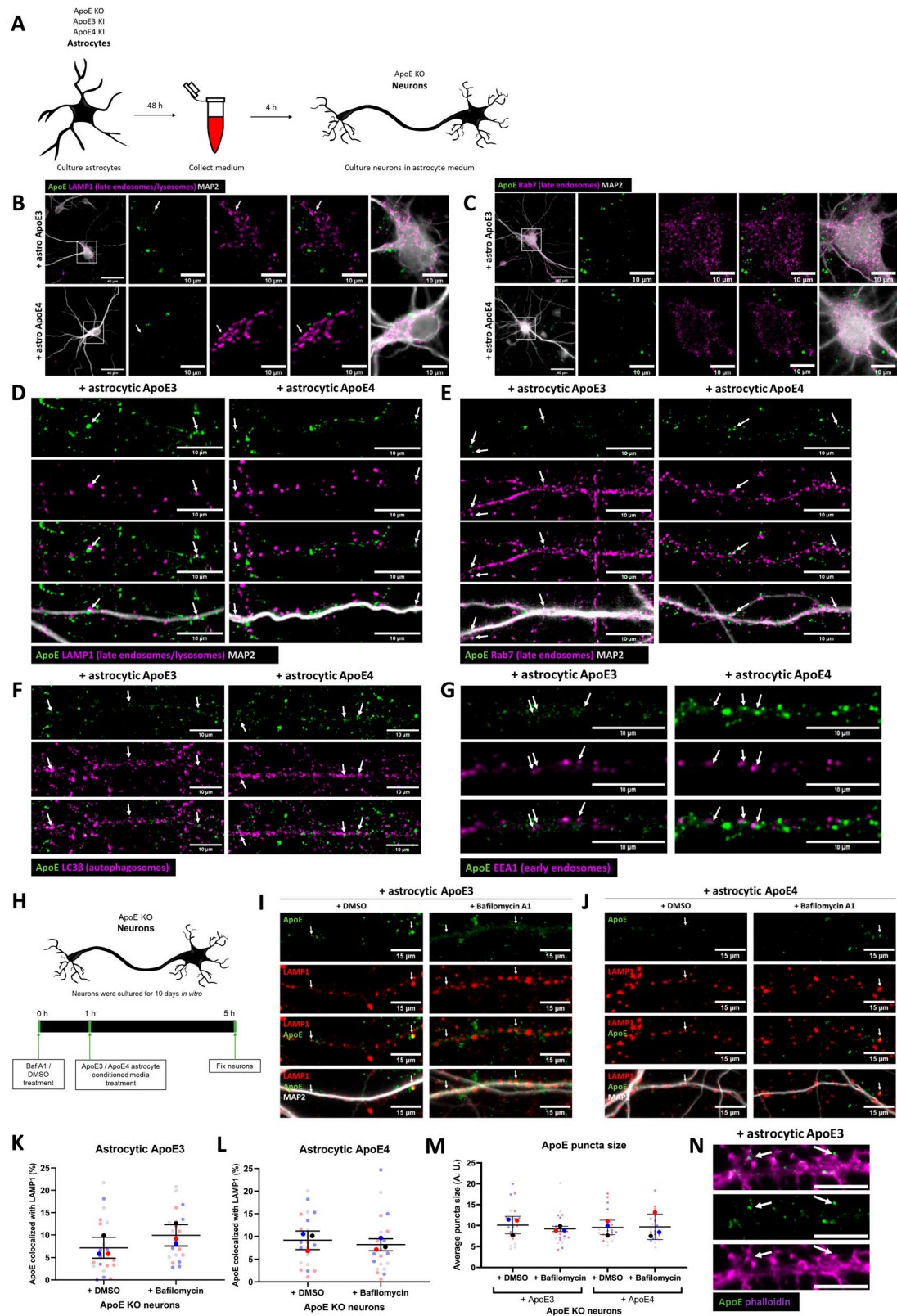

**Figure 2. Internalized ApoE in primary neurons preferentially localize to endosomes and autophagosomes in neurites.**
**(A)** Schematic overview of ApoE KO, ApoE3 KI, and ApoE4 KI astrocyte-conditioned medium collection and subsequent 4 h treatment of ApoE KO primary neuron cultures (19 DIV). **(B, C)** Representative fluorescent images obtained by epifluorescence microscopy of ApoE KO neurons treated with ApoE3 or ApoE4 astrocyte-conditioned medium for 4 h with the focus on neuronal cell bodies. The ApoE astrocyte medium-treated neurons were labeled for ApoE (green), neuronal dendrite marker MAP2 (grey),

after addition of BafA1, suggesting that bafilomycin A1 was successfully blocking lysosomal function in our cultures (Fig S3A) (47). If ApoE is degraded by neuronal lysosomes, it would be expected that the co-localization of ApoE with LAMP1 would increase after bafilomycin A1 treatment. However, no significant change in ApoE–LAMP1 co-localization was detected after BafA1 treatment for either ApoE3 or ApoE4 (Fig 2I–L; median ApoE–LAMP1 co-localization after Baf A1: ApoE3 increased. from 5.2–9.8%, $P$ = 0.0864; ApoE4 decreased from 8.7–6.3%, $P$ = 0.4358). Moreover, no change in ApoE puncta size was detected after Baf A1 treatment in both ApoE3- and ApoE4-treated neurons (Fig 2M, $P$ = 0.7064), suggesting that lysosomal degradation in neurons is not a major pathway of internalized astrocyte-derived ApoE after 4 h of treatment. Although it is expected that internalized ApoE traffics to lysosomes within 4 h, to exclude that ApoE might require longer to reach lysosomes, 1 h of Baf1 A1 was followed by 8 h and 24 h of ApoE3 or ApoE4 co-treatment (Fig S3B and I, respectively). However, ApoE could be hardly detected at these longer time points of Baf A1 treatment (Fig S3C, D, F, G, I, and J). This further supports that ApoE is not degraded in lysosomes in neurons. Interestingly, most remaining ApoE-positive signal in our mixed primary cultures after long-term lysosome inhibition was detected in MAP2-negative cells (Fig S3E, H, and K). Within these cells, ApoE puncta co-localized with LAMP1-positive vesicles (Fig S3E, H, and K, arrows), suggesting astrocyte degradation of ApoE.

Remarkably, many ApoE puncta localized along MAP2-positive dendrites without co-localizing directly with MAP2, pointing towards synaptic localization. In line with our previous work on ApoE isoform effects on synapses (48), astrocyte-derived human ApoE3 and ApoE4 were seen to localize at or near dendritic spines indicated by F-actin marker phalloidin (Fig 2N, arrows).

## Astrocyte-derived ApoE localizes to LAMP1-positive vesicles in primary astrocytes

Although in MAP2-positive neurons some co-localization between internalized ApoE and LAMP1 was seen particularly in their neurites, we also observed that bright ApoE-positive puncta overlapped with LAMP1-positive vesicles in cells negative for the neuronal marker MAP2 (Fig 3A and B). Interestingly, in these MAP2-negative cells,

ApoE particularly co-localized with LAMP1 even after long-term Baf A1 treatment (Fig S3E, H, and K), suggesting that these MAP2-negative cells are degrading ApoE. Because MAP2 labels dendrites, antibody SMI-130 was used to assess whether this added human ApoE might be in axons. However, these strong ApoE-positive puncta did not clearly follow SMI-130 axonal labeling (Fig S4A) and therefore appeared not to be in axons.

To assess the presence of other cell types in our primary mouse brain cultures, which might relate to this strong extra-neuronal ApoE/LAMP1 labeling, the number of MAP2-positive neurons and GFAP-positive astrocytes were initially analyzed in relation to the DAPI nuclei present. 61% (4%) of all cells present in our cultures were positive for MAP2, indicative of neurons (Fig 3C and D). A minority of the cells in our cultures (8% ± 2%) labeled positive for GFAP, suggesting a minority of the cell population represents astrocytes. However, 30% (±3%) of all DAPI-nuclei were negative for MAP2 and GFAP. To further study other cell types, we labeled our WT primary brain cultures with antibodies to Iba1, to identify microglia, and CD140a, to label oligodendrocyte precursor cells (OPCs). As expected in embryonic brain cell cultures, no Iba1-positive microglia were detected (Fig S4B). In contrast, there were numerous CD140a-positive OPCs (Fig S4C). To study whether the ApoE is internalized by astrocytes or OPCs, ApoE KO-mixed brain cultures were treated with ApoE3 and ApoE4 astrocyte-conditioned media and co-labeled for human ApoE and astrocyte marker S100$\beta$ or OPC marker CD140a. Whereas added astrocytic ApoE was not clearly observed in CD140a-positive OPCs (Fig S4D), added human ApoE clearly localized to S100$\beta$-positive astrocytes (Fig 3E) in a similar pattern as seen for LAMP1 (Fig 3B). Thus, primary astrocytes present in our ApoE KO mouse brain cultures readily take up added astrocyte-derived human ApoE where it is detected prominently in LAMP1 vesicles. Interestingly, similar to neurons, internalized ApoE puncta were mostly detected in astrocytic processes rather than cell soma (Fig 4E).

## Internalized ApoE co-localizes with endogenous APP cleavage products in N2a cells and neurons

Previous research supports that intracellular A$\beta$ is generated (21) and in AD accumulates (18) in the endosomal system, which

and either LAMP1 (B) or Rab7 (C) (both shown in magenta). The left panels (B, C) show an overview of the entire neuron; higher magnification images are shown in the other four panels. The white arrows indicate ApoE puncta overlap with LAMP1-positive vesicles in neuronal cell bodies (B). Left panels: scale bars are 40 $\mu$m. Right panels: scale bars are 10 $\mu$m. **(D, E, F, G)** Representative epifluorescence images of neurites from ApoE3 and ApoE4 media-treated primary neurons. **(D, E, F, G)** The neurites are labeled with ApoE (green), MAP2 (D, E) (grey), and late endosomal/lysosomal marker LAMP1 (D), late endosomal marker Rab7 (E), autophagosome marker LC3$\beta$ (F) or early endosomal marker EEA1 (G) (all in magenta). **(D, E, F, G)** White arrows indicate co-localization between ApoE and the subcellular markers (D, E, F, G). Scale bars represent 10 $\mu$m. **(H)** Schematic representation of the lysosome degradation inhibitor Baf A1 and astrocytic ApoE treatment of ApoE KO primary neurons (19 DIV). **(H, I, J)** Representative fluorescence images of ApoE KO neurites treated with control (DMSO) or 10 nM lysosomal inhibitor bafilomycin A1 for 1 h, followed by 4 h ApoE3 (H) or ApoE4 (I) astrocyte-conditioned media. The neurites were labeled for ApoE (green), LAMP1 (red) and MAP2 (grey). ApoE, and LAMP1 co-localization, indicating the presence of ApoE at late endosomes and/or autophagosomes, is indicated by white arrows. Scale bar is 15 $\mu$m. **(K, L)** Quantification of ApoE and LAMP1 co-localization in ApoE3-treated neurons shown in Fig 2I (K) and ApoE4-treated neurons shown in Fig 2J (L) with and without bafilomycin A1 treatment. The researcher performing the quantifications was blinded. The percentages of ApoE co-localizing to LAMP1-positive pixels increased from 5.2–9.8% (Mann–Whitney test, $P$ = 0.0864) in ApoE3-treated and decreased from 8.7–6.3% (Mann–Whitney test, $P$ = 0.4358) in ApoE4-treated neurons after Baf A1 treatment (compared with DMSO control), although these changes were not statistically significant; number of cultures: 3; number of neurons analyzed within each culture: 7; for each neuron, 10 neurites were analyzed and averaged. **(M)** Quantification of the puncta size of added astrocytic ApoE in neurites with and without Baf A1 treatment. No significant difference in ApoE puncta size was observed after Baf A1 treatment in ApoE3- and ApoE4-treated neurons (mean ApoE puncta size: ApoE3 + DMSO: 10.3 pixels, ApoE3 + Baf A1: 9.0 pixels, ApoE4 + DMSO: 9.6 pixels, and ApoE4 + Baf A1: 9.4 pixels) (Kruskal–Wallis test, $P$ = 0.7064); number of cultures: 3; number of neurons analyzed within each culture: 6–7; for each neuron, 10 neurites were analyzed and averaged. Data are shown as mean ± SD. **(N)** Representative fluorescence images showing human ApoE (green) are localized at or close to dendritic spines (indicated by a white arrow) labeled by phalloidin (magenta) in neurites treated with ApoE3 astrocyte-conditioned media for 4 h. Scale bar is 6 $\mu$m. See also Figs S1D and E.

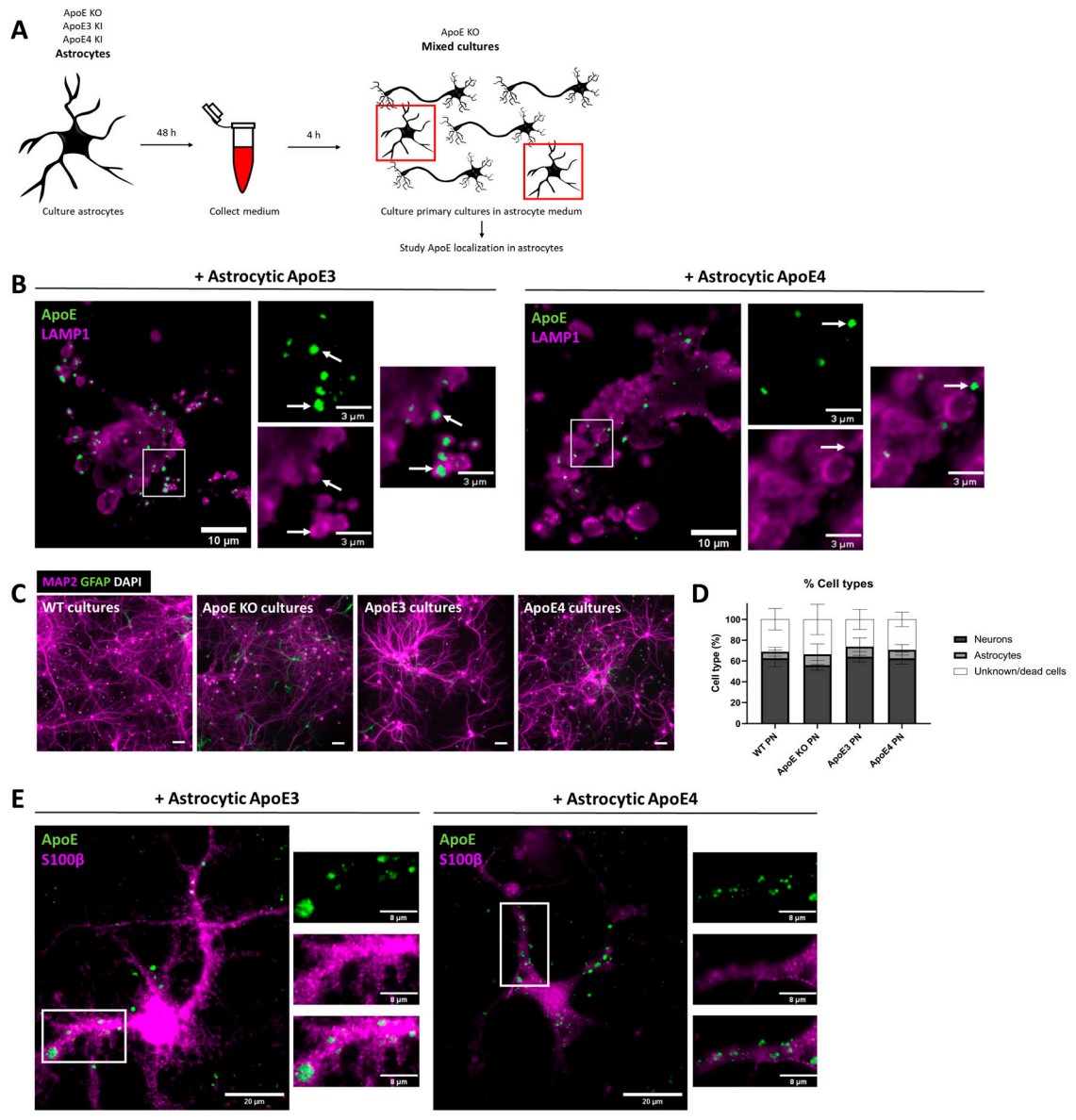

**Figure 3. Astrocyte-derived ApoE localizes to LAMP1-positive vesicle in astrocytes.**
**(A)** Schematic representation of the study design. In brief, conditioned media were obtained from ApoE KO, ApoE3 KI, and ApoE4 KI primary astrocytes and used to treat ApoE KO cultures. Although the majority of cells in our cultures are neurons, astrocytes and other cell types are also present in our mixed cultures. The focus of this figure is on MAP2-negative and GFAP- or S100$\beta$-positive astrocytes. **(B)** Representative fluorescence images of LAMP1-positive puncta (magenta) and ApoE (green) overlap in MAP2-negative cells in primary cultures. The primary brain cultures were treated with ApoE3 or ApoE4 astrocyte-conditioned media for 4 h. The overlap of LAMP1 with human ApoE is highlighted by white arrows. Scale bars are 10 $\mu$m (bigger panels) and 3 $\mu$m (smaller panels). **(C)** Representative images of WT, ApoE KO, ApoE3 KI, and ApoE4 KI primary cultures labeled for neuronal marker MAP2 (magenta), astrocyte marker GFAP (green), and nuclear marker DAPI (white). Scale bar is 50 $\mu$m. **(D)** Quantifications of MAP2-positive and GFAP-positive cells (%) in the primary cultures described in Fig 3C. The total number of cells was set based on the number of DAPI-positive nuclei present. Among the analyzed cultures, on average, 61.3% ± 3.7% of the cells were neurons, 8.7% ± 1.9% were astrocytes, and 30.1% ± 3.1% were GFAP- and MAP2- negative cells (number of cultures: WT: 3; ApoE KO: 2; ApoE3: 4; ApoE4: 4). Data are shown as mean ± SD. **(E)** Representative fluorescence images of S100$\beta$-positive astrocytes (magenta) labeled for ApoE (green). The primary cultures containing the S100$\beta$-positive astrocytes were treated with ApoE3 or ApoE4 astrocyte-conditioned media for 4 h. Scale bar represents 20 $\mu$m. WT, wild-type, PN, primary neurons.

internalized ApoE also accesses. To study whether internalized astrocyte-derived human ApoE and APP processing products intersect at an intracellular level, the cellular localization of human ApoE3 and ApoE4 was first examined in N2a cells overexpressing human APP containing the Swedish mutation (N2a APP$_{Swe}$), which have abundant human APP/A$\beta$ due to overexpression. N2a APP$_{Swe}$

cells were treated with ApoE KO, ApoE3 or ApoE4 astrocyte-conditioned medium for 4 h (Fig 4A) and subsequently co-labeled for human ApoE and antibody 82e1 directed at the N-terminus of human APP cleavage products A$\beta$ and APP $\beta$-C-terminal fragment (APP-$\beta$CTF, also known as C99). Strong co-localization of ApoE and A$\beta$/APP-$\beta$CTF puncta were observed in

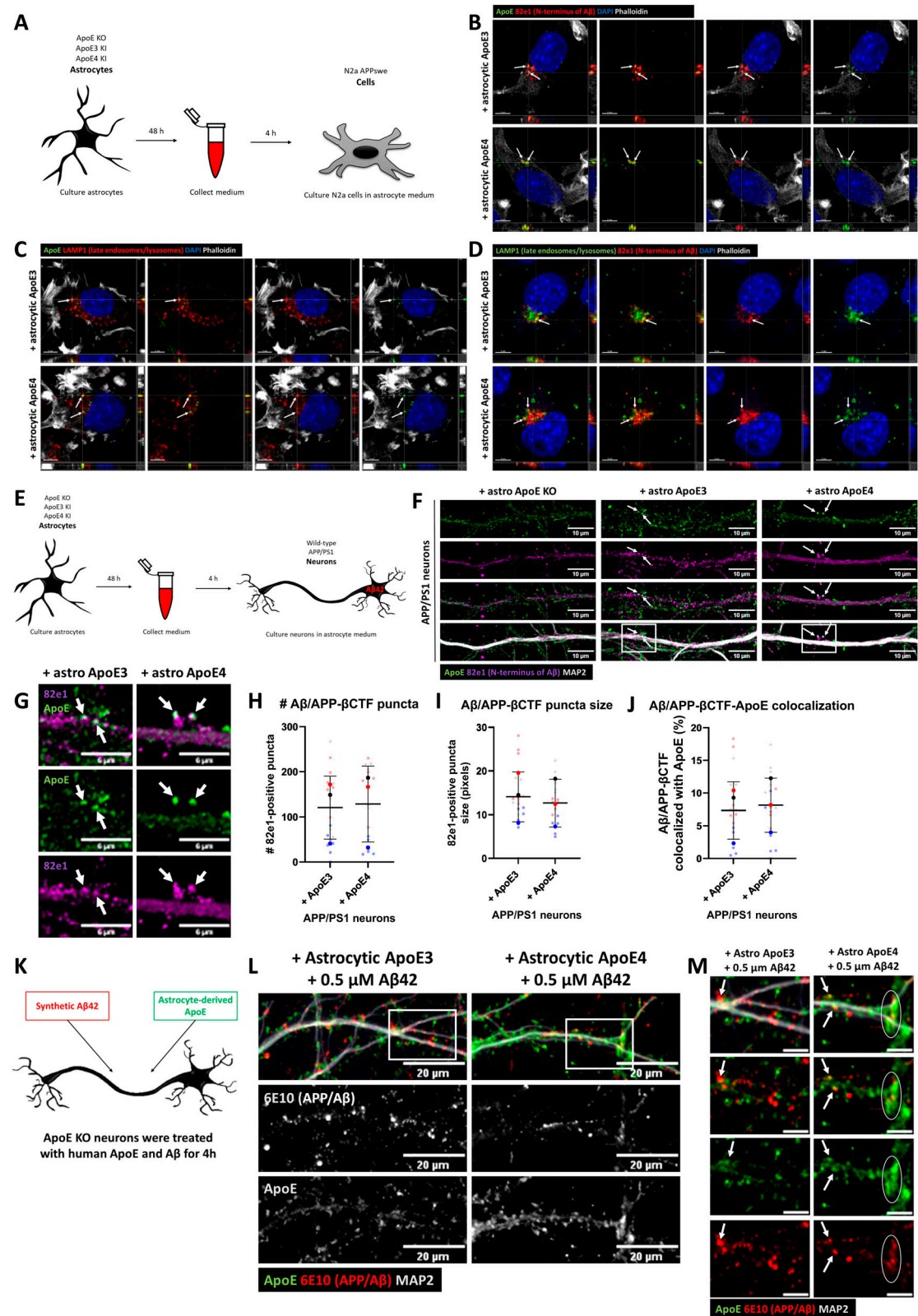

**Figure 4. Internalized ApoE co-localizes with APP cleavage products in neurons and N2a cells.**
**(A)** Schematic representation of experimental approach used. An astrocyte-conditioned medium was collected from ApoE KO, ApoE3 KI, and ApoE4 KI primary astrocytes and subsequently added to N2a APP$_{Swe}$ cells for 4 h. **(B)** Representative orthogonal images obtained by confocal microscopy of N2a APP$_{Swe}$ cells incubated with ApoE3 and ApoE4 astrocyte-conditioned media for 4 h. The cells were labeled for ApoE (green), human Aβ/APP-βCTFs by antibody 82e1 (red), DAPI (blue), and phalloidin (grey). Human

N2a APP$_{Swe}$ cells treated with astrocyte-derived human ApoE3 and ApoE4 (Fig 4B, arrows), but not in untransfected N2a cells or in APP$_{Swe}$ cells treated with ApoE KO-conditioned media (Fig S5A), supporting the conclusion that internalized human ApoE and endogenously produced APP metabolites Aβ/APP-βCTFs intersect intracellularly.

Astrocytic ApoE was similarly internalized into LAMP1-positive late endosomes/lysosomes in untransfected N2a cells (Fig 1G) and N2a APPSwe cells (Fig 4C, arrows), suggesting that the overexpression of human APP with the Swedish mutation in N2a cells does not alter the subcellular localization of human ApoE. To examine whether late endosomes/lysosomes are the subcellular compartments where human ApoE and Aβ/APP-βCTFs intersect, the cellular localization of antibody 82E1 Aβ/APP-βCTF-positive puncta was determined in N2a APPSwe cells. APP-βCTF/Aβ puncta were seen to overlap with LAMP1-positive puncta (Fig 4D, arrows). Altogether, these data suggest that internalized astrocytic human ApoE intersects with APP metabolites Aβ/APP-βCTFs in late endosomal and/or lysosomal compartments in N2a cells.

To study whether ApoE and antibody 82e1-positive APP cleavage products Aβ/APP-βCTFs also co-localize in primary neurons, APP/PS1 transgenic neurons were treated with ApoE KO, ApoE3 or ApoE4 astrocyte media (Fig 4E). Remarkably, ApoE and Aβ/APP-βCTF-positive puncta were also localized together in human ApoE-treated APP/PS1 neurons (Figs 4F and G and S5B), indicating that added ApoE and endogenous Aβ/APP-βCTFs also intersect in primary neurons. The number and size of endogenous human 82e1 antibody-positive Aβ/APP-βCTF puncta were not altered by ApoE isoform (Fig 4H and I; ApoE3-treated compared with ApoE4 82e1 puncta number: $P$ = 0.5427; 82e1 puncta size: $P$ = 0.5708). No difference in ApoE -82e1 co-localization was detected in astrocytic ApoE3- and ApoE4-treated APP/PS1 neurons (Fig 4J; about 8% of 82e1-positive pixels co-localized with ApoE pixels, $P$ = 0.6382), suggesting that internalized astrocytic ApoE3 and ApoE4 co-localized with Aβ/APP-βCTF to a similar extent.

To further examine whether astrocytic ApoE intersects with Aβ in neurons, ApoE KO primary neurons were treated with both ApoE astrocyte-conditioned media and 0.5 $\mu$M synthetic Aβ$_{42}$ (Fig 4K). After 4 h, ApoE and human Aβ42, labeled using the human-specific Aβ/APP antibody 6E10, both localized to MAP2-positive neurites (Figs 4L and S5C) and co-localized along these neurites (Fig 4M, white arrows). Interestingly, although similar ApoE and Aβ$_{42}$ co-localization was noted after ApoE3 and ApoE4 treatment, larger fluorescent puncta of ApoE-Aβ$_{42}$ were observed in ApoE4-treated neurons (Fig 4M, white circle).

## Astrocyte medium but not ApoE genotype influences APP/Aβ levels in N2a APP$_{Swe}$ cells

Because human ApoE derived from primary astrocytes seems to intersect intracellularly with APP cleavage products Aβ/APP-βCTFs, we next examined by Western blot whether human ApoE affects the levels of APP metabolites and whether it does so in an ApoE isoform-dependent manner in astrocyte-conditioned media–treated N2a APP$_{Swe}$ cells (Fig 5A). Western blot showed that human APP and Aβ protein levels in N2a APP$_{Swe}$ cells and their media were not altered by the presence of the different ApoE astrocyte media after 15 min incubation (Fig S6A–D). After 4 h treatment, the time point where we observed internalized ApoE and Aβ/APP-βCTF co-localization (Fig 4B), APP and Aβ protein levels were also not altered in both N2a APP$_{Swe}$ lysate (APP: $P$ = 0.5281; Aβ: $P$ = 0.8427) and media (sAPPα: $P$ = 0.9370; Aβ: $P$ = 0.5861) as assessed by Western blot (Fig 5B–D and G–I, blue panel; Fig S6E and F). However, we noted using Western blot that full-length APP protein levels were significantly increased ($P$ = 0.0162), whereas secreted sAPPα levels were significantly decreased ($P$ = 0.0009) in N2a cells after 8 h astrocyte media treatment, independently of the presence of human ApoE (Fig 5B, E, G, and J, orange panel; Fig S6E and F), because the addition of APOE KO astrocyte medium also induced this effect. This suggests that the presence of astrocyte-conditioned media itself, and not human ApoE, increases APP protein expression in N2a APP$_{Swe}$ cells. ApoE3 and ApoE4 astrocyte-conditioned media treatment did not significantly alter intracellular Aβ protein levels in N2a APP$_{Swe}$ cells (Fig 5B and F; $P$ = 0.7347), whereas secreted Aβ levels were

---

ApoE and 82e1-positive Aβ/APP-βCTFs overlap as indicated by white arrows. **(C)** Orthogonal fluorescent images of astrocytic ApoE3 and ApoE4-treated N2a APP$_{Swe}$ cells. ApoE was labeled in green, late endosomes/lysosomes with LAMP1 in red, DAPI in blue, and phalloidin in grey. Arrows indicate overlap between ApoE- and LAMP1-positive puncta. **(D)** Representative orthogonal confocal images of N2a APP$_{Swe}$ cells treated with astrocyte-derived ApoE3 and ApoE4 for 4 h. The cells are labeled for LAMP1-positive vesicles (green), 82e1-positive Aβ/APP-βCTFs (red), and DAPI (blue) to study the localization of Aβ-containing APP-processing products in N2a APP$_{Swe}$ cells. White arrows highlight 82e1-positive puncta co-localizing with LAMP1-positive puncta. Scale bars are 5 $\mu$m (B, C, D). **(E)** Schematic overview of ApoE KO, ApoE3 KI, and ApoE4 KI astrocyte-conditioned media collection and 4 h treatment of WT and AD APP/PS1 transgenic primary neurons (19 DIV). **(F)** Representative confocal images of APP/PS1 neurons incubated with ApoE KO, ApoE3 or ApoE4 astrocyte-conditioned media for 4 h. The intracellular intersection between internalized ApoE and endogenous human APP cleavage products was studied by labeling neurites for ApoE (green), APP metabolites Aβ/APP-βCTFs (82e1) (magenta), and MAP2 (grey). Arrows indicate ApoE and antibody 82e1 Aβ/APP-βCTF co-localization at neurites. Scale bar is 15 $\mu$m. **(G)** Higher magnification images of Fig 4F (indicated by a white box) of ApoE3 and ApoE4 astrocyte-conditioned media–treated APP/PS1 neurons. The white arrows point at subcellular co-localization of internalized astrocyte-derived ApoE with endogenous human Aβ/APP-βCTFs. Scale bar is 6 $\mu$m. **(H, I)** Quantification of 82e1-positive Aβ/APP-βCTF puncta number (H) and size (I) in astrocytic ApoE3- and ApoE4-treated APP/PS1 primary neurons. No significant difference in Aβ/APP-βCTF puncta number and size were detected between the ApoE isoform treatments (82e1 puncta number: Mann–Whitney, $P$ = 0.5427; median ApoE3 treatment: 99, median ApoE4 treatment: 152) (82e1 puncta size: $t$ test, $P$ = 0.5708; mean for ApoE3 treatment: 13.7, mean for ApoE4 treatment: 12.6). The number of embryos (big data points in the graph): 3, number of neurons per embryos (small data points in the graph): 6–7 (10 neurites per neuron were analyzed and averaged per neuron). **(J)** Quantification of 82e1-positive Aβ/APP-βCTF and ApoE co-localization in ApoE3- and ApoE4-treated APP/PS1 neurons. Astrocytic ApoE3 and ApoE4 co-localized with 82e1-positive Aβ/APP-βCTF to a similar extent (mean ApoE3-82e1 co-localization: 7.3%, mean ApoE4-82e1 co-localization: 8.1%; $t$ test, $P$ = 0.6382). The number of embryos (big data points in the graph): 3, number of neurons per embryos (small data points in the graph): 6–7 (10 neurites per neuron were analyzed and averaged per neuron). **(K)** Schematic representation of synthetic 0.5 $\mu$M human Aβ$_{42}$ and astrocyte-derived ApoE3 or ApoE4 treatment of neurons that do not endogenously produce ApoE (ApoE KO). **(L)** Representative confocal microscopy images of Aβ$_{42}$ and astrocytic ApoE double-treated primary neurons. The neurons were labeled for ApoE, APP/Aβ (6E10), and MAP2. Scale bar represents 20 $\mu$m. **(M)** High-magnification images of Aβ$_{42}$ and ApoE-treated neurons of the area shown by a white box in Fig 4L. White arrows indicate ApoE and APP/Aβ overlap; the white circle highlights overlap of larger puncta of ApoE and APP/Aβ. Scale bar is equal to 6 $\mu$m. See also Fig S5A and B.

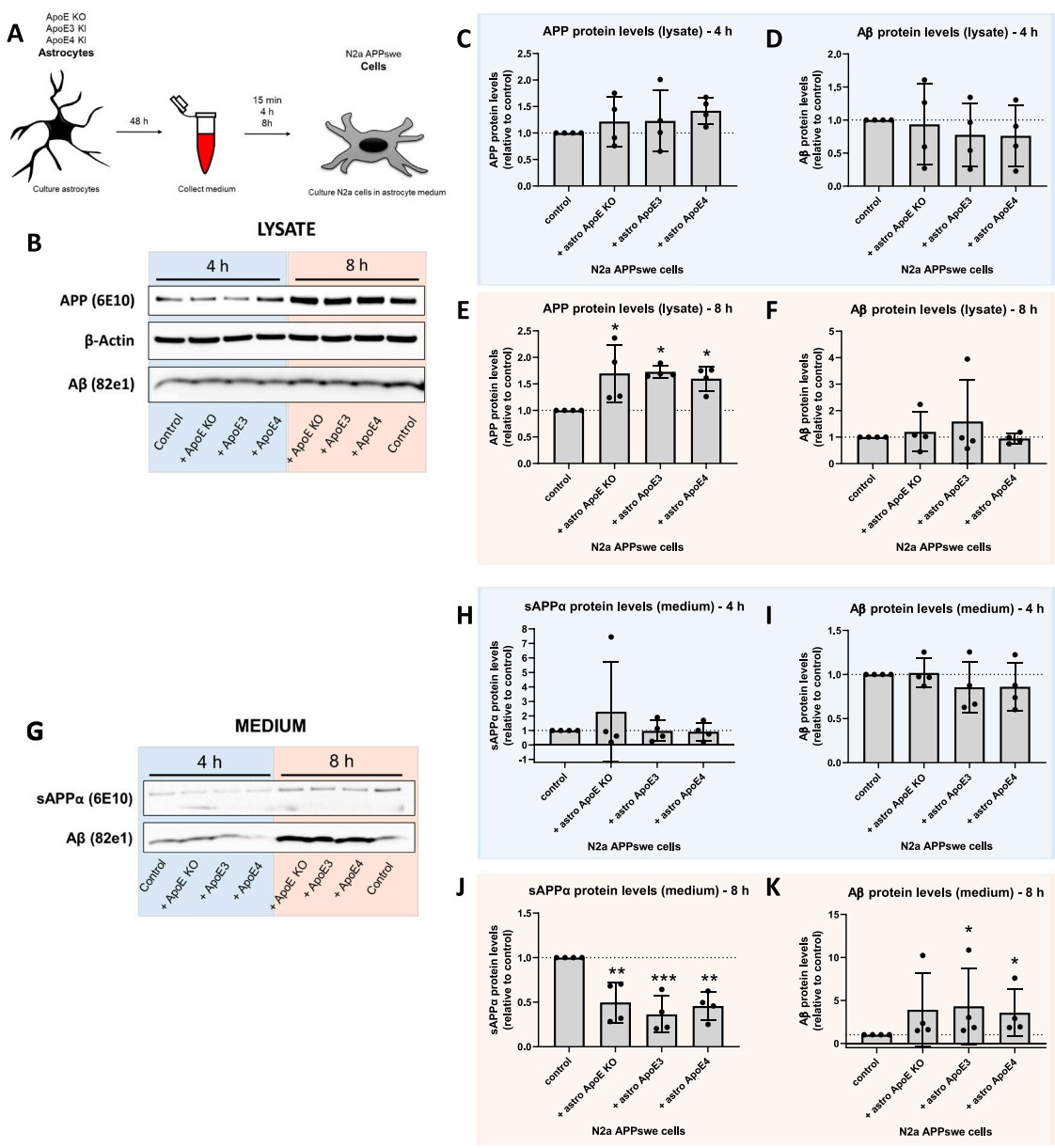

**Figure 5. Astrocyte media but not the ApoE genotype influence APP/Aβ levels in N2a APP_Swe cells.**
**(A)** Schematic overview of conditioned media treatment from ApoE KO, ApoE3-KI, and ApoE4-KI astrocytes to N2a APP_Swe cells for 15 min, 4 h or 8 h. **(B)** Representative Western blot bands of lysate of N2a APP_Swe cells treated with ApoE KO, ApoE3 or ApoE4 astrocyte-conditioned media or control (fresh N2a media) for 4 h (blue) and 8 h (orange). The Western blot membranes were stained for APP, detected by 6E10 antibody, β-actin, and Aβ, detected using antibody 82e1. **(C, D, E, F)** Quantification of Western blots shown in Fig 5B. Full-length APP and Aβ protein levels were quantified in lysates of N2a APP_Swe cells treated with astrocyte-conditioned media for 4 h ((C, D), respectively, blue) or 8 h ((E, F), respectively, orange). No differences in APP (4 h (C); One-way ANOVA, $P$ = 0.5281) and Aβ protein levels (4 h (D): One-way ANOVA, $P$ = 0.8427; and 8 h (F): Kruskal–Wallis test, $P$ = 0.8427) were detected after 4 h and 8 h of astrocyte media treatment in the lysates of N2a APP_Swe cells. After 8 h, APP levels in the lysates were significantly up-regulated by addition of astrocyte-conditioned media ((E), One-way ANOVA, control versus ApoE KO media: $P$ = 0.0171; control versus ApoE3 media: $P$ = 0.0134; control versus ApoE4 media: $P$ = 0.0404). Values were normalized to β-actin and control N2a APP_Swe cells (APP and Aβ protein levels in these cells were set to 1). Data are shown as mean ± SD. **(G)** Representative Western blot membranes of sAPPα and Aβ protein levels in media from N2a APP_Swe cells treated with ApoE KO, ApoE3 or ApoE4 astrocyte media or control (fresh N2a media) for 4 h (blue) or 8 h (orange). **(H, I, J, K)** Quantification of secreted sAPPα and Aβ protein levels in the conditions shown in Fig 5G. sAPPα and Aβ levels were determined in N2a APP_Swe cells treated with astrocyte media for 4 h ((H, I), respectively, blue) or 8 h ((J, K), respectively, orange). **(H, I)** After 4 h, astrocyte-conditioned media treatment, no differences in sAPPα and Aβ levels were detected in N2a APP_Swe media (sAPPα (H): Kruskal–Wallis test, $P$ = 0.9370; Aβ (I): One-way ANOVA, $P$ = 0.5861). **(J)** After 8 h, sAPPα protein levels in N2a media were increased by all astrocyte media treatments ((J), One way ANOVA, control versus ApoE KO media: $P$ = 0.0036; control versus ApoE3 media: $P$ = 0.0006; control versus ApoE4 media: $P$ = 0.0020). **(K)** Astrocytic ApoE3 and ApoE4 media, but not ApoE KO media, also increased Aβ levels in media after 8 h treatment ((K), Kruskal–Wallis test, control versus ApoE KO media: $P$ = 0.1086; control versus ApoE3: $P$ = 0.0329; control versus ApoE4: $P$ = 0.0329). Data are shown as mean ± SD. * $P$-value < 0.05, ** $P$-value < 0.01. For all experiments, the number of embryos used is N = 4 (for all groups). See also Fig S6.

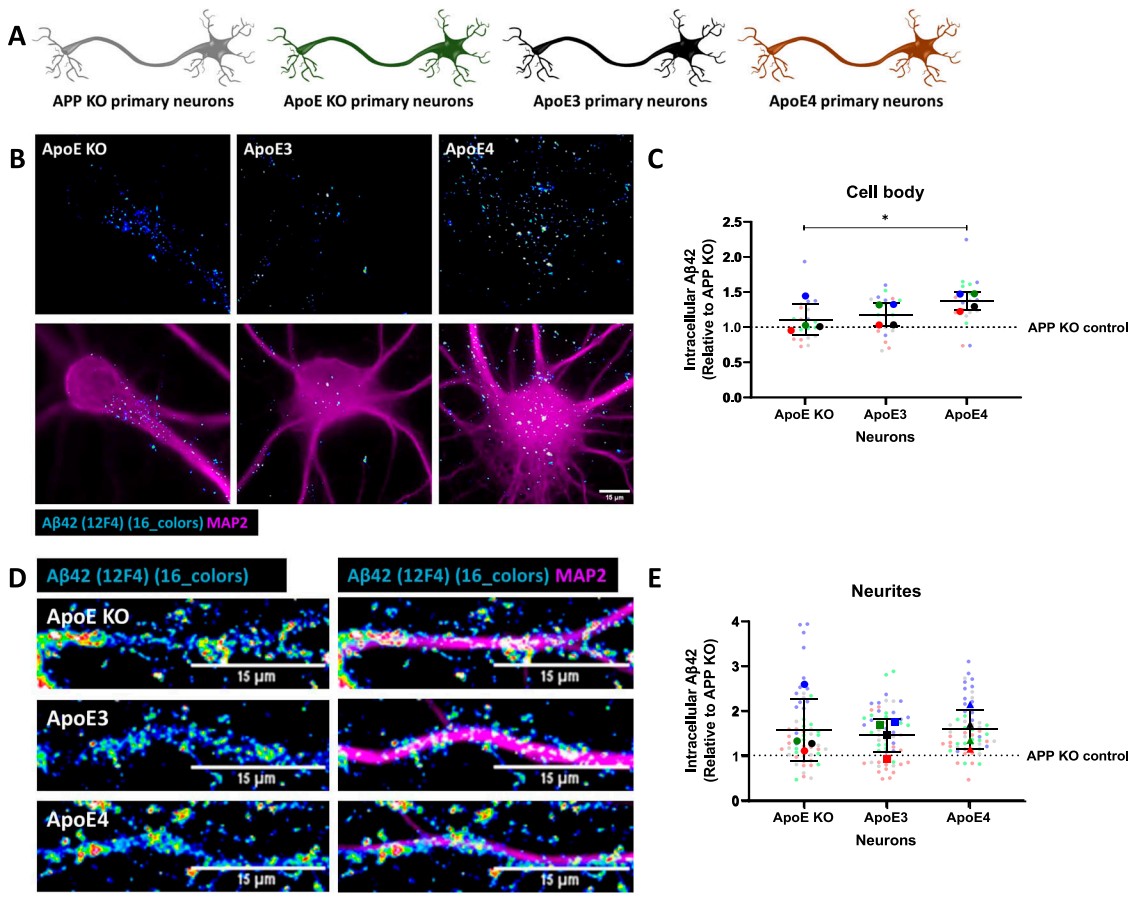

**Figure 6.   Increased endogenous intraneuronal Aβ$_{42}$ levels in cell bodies of cultured ApoE4 neurons.**
**(A)** Schematic visualization of the different primary neuron models used to study the effects of human ApoE isoforms on levels of intraneuronal Aβ$_{42}$. **(B)** Representative epifluorescence images of neuronal cell bodies from ApoE KO, ApoE3-KI, and ApoE4-KI primary neurons (19 DIV). The neurons were labeled for endogenous mouse Aβ$_{42}$ using antibody 12F4 (16 colors) and MAP2 (magenta). Scale bar is 15 μm. **(C)** Quantification of endogenous Aβ$_{42}$ in neuronal cell bodies as shown in Fig 6B. The levels of Aβ$_{42}$ were significantly higher in ApoE4-KI neurons compared with ApoE KO neurons (23.3% increase in ApoE4 compared with ApoE KO neurons, Kruskal–Wallis test, P = 0.0146). Number of cultures: 4 (big data points in the graph), number of cell bodies analyzed per culture: 5 (small data points in graph). **(D)** Representative images of ApoE KO, ApoE3-KI, and ApoE4-KI neurites labeled for Aβ$_{42}$ by antibody 12F4 (16 colors) and MAP2 (magenta). Scale bar is equal to 15 μm. **(E)** Quantification of intracellular Aβ$_{42}$ levels, as measured by antibody 12F4 intensity, of neurites of ApoE KO, ApoE3-KI, and ApoE4-KI primary neurons. No significant differences were found in 12F4 intensity between the neurites of ApoE KO, ApoE3, and ApoE4 neurons (Kruskal–Wallis test, P = 0.5960). Number of cultures per condition: 4 (big data points in graph), number of neurites analyzed per culture: 15 (small data points in graph). Data are shown as mean ± SD. The dashed line indicates the level of unspecific signals detected in APP KO control neurons. * P-value < 0.05. See also Fig S5D and E.

significantly increased (Fig 5G and K; 0.0151); a trend for elevated secreted Aβ level was also noted at 8 h with ApoE KO astrocyte media (P-value: 0.1086) (Fig 5K).

### ApoE4 induces increased intraneuronal Aβ$_{42}$ levels in cultured neurons

To study whether human ApoE alters Aβ levels in primary neurons, Aβ$_{40}$ and Aβ$_{42}$ levels were determined in media collected from primary ApoE KO brain cultures treated with ApoE3 or ApoE4-astrocyte-conditioned media for 24 h. To detect mouse Aβ in the media, the more sensitive technique of mesoscale analysis was used. Interestingly, treating neurons with astrocyte-conditioned media increased Aβ$_{42}$ but not Aβ$_{40}$ levels in neuron-conditioned media in an ApoE-independent manner (Fig S7A and B) and increased the Aβ$_{42}$/Aβ$_{40}$ ratio secreted by neurons (Fig S7C).

Endogenous mouse Aβ$_{42}$ was next investigated by immunofluorescence using Aβ$_{42}$ antibody 12F4, as described previously (10), in primary brain cultures obtained from ApoE KO, ApoE3-KI, and ApoE4-KI mice (Fig 6A). Interestingly, ApoE4 neurons showed significantly higher intensity of Aβ$_{42}$ labeling in neuronal cell bodies compared with ApoE KO neurons (Figs 6B and C and S5D; 23.3% increase, P = 0.0174), although no statistical difference in Aβ$_{42}$ signal was detected between the different ApoE isoforms in neurites (Figs 6D and E and S5E, P = 0.5960).

To further examine isoform-dependent effects on intraneuronal Aβ, we treated primary human ApoE KI mouse brain cultures that endogenously express human ApoE3 or ApoE4 in both neurons and astrocytes with synthetic human Aβ$_{42}$ for 4 h (Fig 7A). Of note, the number of internalized human Aβ$_{42}$ puncta, detected using the human-specific Aβ/APP antibody (6E10) were increased by 51.7% and 60.4% in ApoE4 compared with ApoE KO and ApoE3 primary

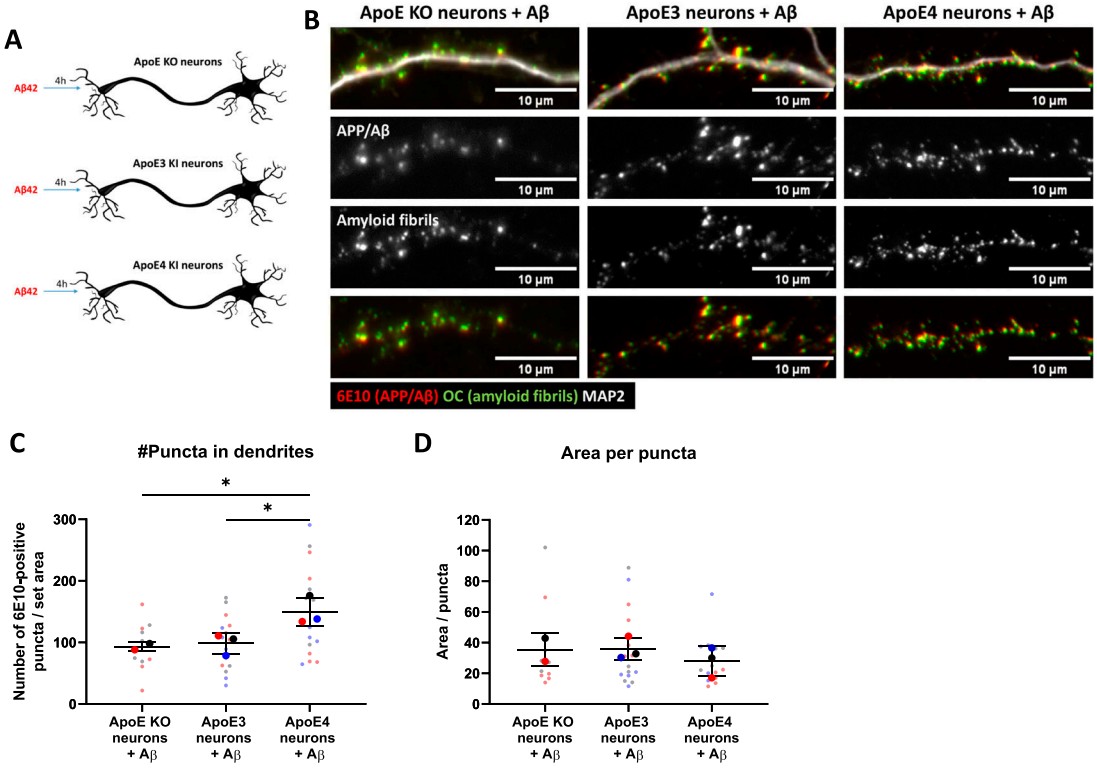

**Figure 7. Internalized Aβ₄₂ levels are increased in human ApoE4-KI primary neurons.**
**(A)** Schematic representation of 0.5 μM synthetic Aβ₄₂ treatment of different ApoE neuron cultures: ApoE KO, and human ApoE3- and ApoE4-KI neurons.
**(B)** Representative epifluorescence images of neurites from Aβ₄₂-treated ApoE KO, ApoE3, and ApoE4 primary neurons. The neurites are labeled for human Aβ with antibody 6E10 (red), fibrillar oligomer antibody OC (green), and MAP2 (grey in upper panel). Clear co-localization was observed between 6E10 and OC, suggesting that added synthetic Aβ₄₂ is aggregated inside neurons. **(C, D)** Quantification of the number of (C) and the average area size of the 6E10-positive Aβ₄₂ puncta (D) in Aβ₄₂-treated primary neurons as shown in Fig 7B. The number of antibody 6E10 puncta is significantly increased in ApoE4 compared with ApoE KO (51.8% increase by ApoE4, one-way ANOVA, $P = 0.0474$) and ApoE3 (60.4% increase by ApoE4, one-way ANOVA, $P = 0.0453$) neurons after Aβ₄₂ treatment. Number of cultures analyzed: ApoE KO: 2, ApoE3: 3, ApoE4: 3; number of neurons analyzed per culture: 5 (per neuron, 10 neurites were analyzed and averaged). The quantifications were performed while being blinded (C, D). Data are shown as mean ± SD. * < 0.05. px, pixels. See also Fig S5F.

brain cultures, respectively (Figs 7B and C and S5F; ApoE KO versus ApoE4: $P = 0.453$, ApoE3 versus ApoE4; $P = 0.474$). No difference in the area of 6E10-positive Aβ₄₂ puncta was detected (Fig 7B and D, $P = 0.5568$). Strikingly, almost all the intraneuronal human Aβ puncta in both ApoE3 and ApoE4 neurons were also detected by the antibody OC against Aβ fibrils and oligomeric fibrils (Fig 7B, lowest panel), supporting that the internalized human Aβ₄₂ is aggregated (27).

## Discussion

The prior literature suggests that ApoE4 plays a role in endosomal (dys)function (7, 12, 14, 16, 49). However, intracellular trafficking of ApoE in normal and AD conditions remain poorly studied. We now show that ApoE is present in the endosome–autophagy–lysosome system of N2a neuroblastoma cells and in primary neurons and astrocytes, albeit in somewhat different subcellular patterns, with neurons showing endosome–autophagosome labeling particularly of neurites and less in lysosomes. We demonstrate for the first time that astrocytic human ApoE is internalized and intersects with

endogenous APP cleavage products Aβ/βCTF in primary neurons and N2a APP_Swe cells. ApoE3 compared to ApoE4 was not seen to differentially alter endogenous APP in culture, although astrocyte media, even when devoid of ApoE, increased APP levels. However, ApoE4 increased endogenous intraneuronal mouse Aβ₄₂ levels in primary neurons compared with the absence of ApoE. In addition, ApoE4 altered intraneuronal levels of internalized exogenously added Aβ₄₂ in an ApoE isoform-dependent manner (ApoE4 > ApoE3). Our study highlights the importance of the endosome–autophagy system in ApoE biology and reveals an intersection between astrocyte-derived ApoE and APP processing products in neurites that we hypothesize to be of considerable importance in the pathogenesis of AD.

Because of the highly polarized shape of neurons, neuronal endolysosomal trafficking is quite distinct from other cell types, which might relate to our observation that ApoE clearly localized to lysosomes in N2a cells but not in neurons. In neurons, lysosomal maturation occurs during retrograde transport of endosomes–autophagosomes in neuronal processes towards the cell soma, with the most mature lysosomes being present in the soma itself (40, 41). The fact that we did not observe ApoE in lysosomes in the

neuron cell soma, even after BafA1 treatment, suggests that ApoE is not trafficked to and degraded by lysosomes in neurons. Previous studies (12, 13, 50), including a study in primary neurons (12), demonstrated that ApoE is recycled, implicating endosomal recycling and re-secretion as pathways relevant to neuronal ApoE trafficking. Impaired recycling and re-secretion rather than lysosomal degradation were also shown to occur with internalized ApoE4 compared with ApoE3 in primary hepatocytes (13) and in a neuronal cell line (51).

Our study demonstrates that astrocyte-derived human ApoE is present at APP-processing sites, with co-localization of ApoE and human Aβ/APP-βCTF in neurons and neuroblastoma cells. As antibody 82E1 directed against the free N-terminus of Aβ and APP-βCTF does not differentiate between these, we cannot draw conclusions on ApoE specifically intersecting with either Aβ or APP-βCTF. The Aβ domain, however, resides in both Aβ and APP-β-CTFs and both are viewed as participating in the pathogenesis of AD. In N2a cells, ApoE and Aβ/APP-βCTF were detected in LAMP1-positive vesicles, supporting that late endosomes, autophagosomes and/or lysosomes are the cellular sites where ApoE intersects with Aβ/APP-βCTF. In contrast, in neurons, ApoE localized to diverse endosomal–autophagic vesicles but not to lysosomes. However, the endosome–autophagy–lysosome system may differ in neurons in primary culture compared with the brain. On electron micrographs, lysosomes in brain are neither normally localized to axons nor to dendrites other than their most proximal parts (52). Because of the unique endosome–autophagy–lysosome biology of neurons (53), additional work is required to better define the subcellular site(s) and implications of the ApoE and Aβ/APP-βCTF intersection in neurons for AD.

Although our work provides no evidence of direct interaction of ApoE and Aβ/APP-βCTF, it is known that ApoE and Aβ interact in human AD brains (27, 54, 55). Moreover, Kuszyczk et al (27) reported that inhibition of Aβ and ApoE binding reduces intraneuronal Aβ levels and protects against AD-like Aβ-induced synapse alterations. Thus, the intersection of Aβ/APP-βCTF and ApoE we observed in neuronal vesicles likely affects intraneuronal Aβ via direct interactions. Previous articles also reported that the presence of ApoE in general is linked to increased intraneuronal Aβ compared with when ApoE is KO (27, 56). Huang et al (57) reported that human ApoE produced in HEK293T cells influences APP and Aβ secretion in an ApoE isoform-dependent manner (ApoE4 > ApoE3 > ApoE2). Here, we showed that human ApoE derived from primary astrocytes did not induce ApoE-specific effects on the secretion of APP and Aβ nor on endogenous levels of APP in N2a cells or neurons. In fact, we observed that just adding astrocyte-conditioned media even devoid of ApoE to N2a cells increased APP. Our data are consistence with Huang et al, who showed that the ApoE isoform effect on Aβ production was abolished when glia were present in the culture (57). Together, these data support the conclusion that factors in glia media markedly affect APP and Aβ levels independent of ApoE. A recent study on human iPSC-derived neurons reported an isoform-dependent astrocytic ApoE4 media increase in APP levels (58); in contrast, in vivo studies indicate that ApoE genotype does not affect APP mRNA and protein levels in brains of humanized ApoE target replacement mice (26) and in transgenic PDAPP mice cross-bred with ApoE target replacement

mice (59), arguing against an effect of human ApoE isoforms on APP metabolism. Wang et al (36) recently suggested that lipidated ApoE with cholesterol could change the localization of APP at the membrane in N2a cells towards GM1 lipid clusters, rather than changing its expression levels. As β-secretase is associated to lipid clusters, whereas α-secretase is linked to non-lipid-rich membrane regions, the ApoE-induced APP shift to lipid clusters in the membrane could favor Aβ production without affecting APP levels. It is also possible that the complex homeostatic control of Aβ levels might limit actual changes in levels that can be detected in cellular experiments (60).

We saw a trend but no statistical difference in ApoE3 versus ApoE4 on endogenous levels of $A\beta_{42}$ although we did detect significantly more intraneuronal $A\beta_{42}$ with ApoE4 than ApoE KO-conditioned media (Fig 6). However, when adding elevated levels of $A\beta_{42}$ to neurons, we then did reveal an ApoE isoform-specific effect on intraneuronal $A\beta_{42}$, with ApoE4 inducing an increased number of intraneuronal $A\beta_{42}$ puncta compared to ApoE3 (Fig 7). This suggests that ApoE4 affects $A\beta_{42}$ internalization, subsequent aggregation and/or degradation in neurons. A study in N2a cells further supports this by reporting enhanced Aβ internalization when co-cultured with ApoE4-expressing cells (61). Although the potential role of intraneuronal ApoE and its intersection with Aβ/APP-βCTF requires further study, that ApoE intersects with endogenous APP processing products and added Aβ, points to a role of the ApoE and Aβ intersection at different levels related to APP and Aβ biology.

In addition to the endosome–autophagosome pattern of ApoE labeling that we now describe in neurites, we previously described that added astrocyte-derived ApoE localized to synaptic terminals in primary neurons (48). Although the focus of the current article was on internalized ApoE in neurons, ApoE in the current study was also seen in a pattern of labeling around cell bodies and neurites consistent with the synaptic localization described in our previous article. Koffie et al (33) showed that ApoE co-localizes with Aβ oligomers at synapses in human brains, highlighting synaptic terminals as a potential site of ApoE and Aβ intersection. Endosomal trafficking is crucial for synaptic function (34, 62) and abnormal endosomal regulation, for example, by overexpressing Rab5, causes synaptic dysfunction (63, 64). We previously showed altered neuronal activity dependent on ApoE genotype (48), however, it remains unclear whether and how this might relate to altered endosome–autophagosome trafficking. Next to neurons, we detected ApoE internalization into primary astrocytes where it also appeared more prominent in their processes than cell soma (Fig 3).

In conclusion, due to the complex endolysosomal biology of neurons, the endocytic trafficking of ApoE is different in neurons, as internalized ApoE does not end up in lysosomes of neuronal cell bodies as it does in N2a cells and primary astrocytes. In addition, ApoE and Aβ/APP-βCTF, all key players in AD, intersect subcellularly, with isoform-dependent ApoE effects. Our work further highlights the importance of the endocytic pathway in relation to the major AD players ApoE and Aβ/APP-βCTF (65, 66) in the early cellular phase of AD, and suggests that the endosome–autophagy–lysosome system is a potential site where ApoE and Aβ interact in AD.

# Materials and Methods

## Animals

In this study, the following mouse models were used: ApoE KO (B6.129P2-Apoe<tm1Unc>/J; Jackson Laboratory), humanized ApoE3-KI (B6.Cg-Apoeem2(APOE*)Adiuj/J; Jackson Laboratory), humanized ApoE4-KI (B6(SJL)-Apoetm1.1(APOE*4)Adiuj/J; Jackson Laboratory), and AD transgenic APP/PS1 B6.Cg-Tg mice (PSEN1dE9; APPswe) 85Dbo/Mmjax. All animal experiments described in this study were approved by the Ethical Committee for animal research at Lund University, Sweden (permit number: M5983-19).

## Primary mouse mixed brain cultures

Primary mouse brain cultures were obtained from cortical and hippocampal brain tissues from ApoE KO, ApoE3-KI, ApoE4-KI, WT, and APP/PS1 embryos (E15–E17). Embryonic brain tissue was dissected and dissociated into single cells as previously described by Takahashi et al ([67]). In short, cortical and hippocampal embryonic brain tissues were manually dissociated using 0.25% trypsin (15090046; Thermo Fisher Scientific) and seeded onto poly-D-lysine–coated coverslips or plates. Directly after seeding, the cells were cultured in DMEM containing 10% FBS (10082147; Gibco) and 1% antibiotics penicillin/streptomycin (p/s) (SV30010; Thermo Fisher Scientific). After 3–5 h, the 10% FBS medium was replaced by complete Neurobasal medium containing B27 supplement (17504044; Gibco), 1% p/s, and 1.4 mM L-glutamine (25030081; Gibco) and cultured for 19 days in vitro (DIV) until further use. Both male and female embryos were used to generate primary brain cultures, although we did not specifically identify the sex of embryos; our blinding to gender also could reduce potential bias in subsequent analyses. Our experience indicates that the offspring of our crosses as expected generate about equal numbers of male and female offspring.

## Primary mouse astrocytes

Primary mouse astrocyte cultures were obtained from ApoE KO, ApoE3-KI, and ApoE4-KI mouse pups (P1-P3) as previously described by Konings et al ([48]). In short, cortical and hippocampal brain tissues were obtained from mouse pup brains and manually dissociated after 0.25% trypsin incubation using plastic Pasteur pipets. After dissection and dissociation of the tissue, cells were seeded on T75 plates coated with poly-D-lysine. Primary astrocytes were cultured in AstroMACS medium (130-117-031; Miltenyi Biotec) with 0.5 mM L-glutamine and the medium was replaced every 2–3 d. Analogous to primary brain cultures, also for our generation of primary astrocytes we did not separate cells based on gender, but used all available pups which should lead to analogous number of male and female cells.

## Culturing neuroblastoma N2a cells

Mouse neuro-2a (N2a) cells (ATCC CCL-131) without transfection or N2a cells stably transfected with human APP carrying the Swedish mutation (N2a APP_Swe) ([68]) were cultured in DMEM and Opti-MEM (31985062; Gibco) (ratio 1:1) containing 10% FBS and 1% p/s at 37°C and 5% $CO_2$. N2a cells transfected with human APP_Swe were selected using 50 mg/ml Geneticin (10131027; Gibco) in their media. N2a and N2a APP_Swe cells were seeded on poly-D-lysine coated glass coverslips one day before recombinant or astrocytic ApoE treatment.

## Recombinant ApoE treatment

Recombinant ApoE3 and ApoE4 proteins (SRP4696 and A3234; Sigma-Aldrich, respectively) were reconstituted in 0.1% BSA in milli-Q water to a stock concentration of 0.1 mg/ml. All experiments used a final ApoE concentration of 2.5 μg/ml.

## Astrocyte-conditioned media treatment

Once primary astrocyte cultures reached 80% confluence, ApoE KO, ApoE3, and ApoE4 astrocyte conditioned media were collected. For the medium collection, astrocytes were first shortly washed with PBS and cultured in complete Neurobasal medium. After 48 h, astrocyte conditioned media were collected on ice, centrifuged at 9,279*g* (10,000 rpm) at 4°C for 10 min, and supernatant stored at −80°C in small aliquots to avoid freeze–thaw cycles.

Astrocyte-conditioned media from ApoE KO, ApoE3, and ApoE4 astrocyte cultures were used to treat primary neuron cultures or mouse neuroblastoma neuro-2a (N2a) cells. Half of the original culture medium from primary neurons and N2a cells was replaced by an equal volume of conditioned astrocyte medium, resulting in half astrocyte-conditioned and half Neurobasal/N2a medium.

## Lysosome inhibition experiments

To block lysosomal function in neurons, lysosomal acidification inhibitor bafilomycin A1 (Sigma-Aldrich) was used. Bafilomycin A1 was reconstituted in DMSO and further diluted to a final concentration of 10 nM (5 and 9 h) or 2 nM (25 h) to treat primary neurons. Primary neurons were treated 1 h before ApoE addition for a total duration of 5 h before fixation. For longer time points, ApoE KO primary neurons were treated with bafilomycin A1 1 h before 8 h or 24 h ApoE astrocyte-conditioned media co-treatment.

## Synthetic Aβ_{42} treatment

Synthetic Aβ42 peptides (Cat#1428; Tocris) were prepared as previously described by Klementieva and colleagues ([69]). In short, synthetic Aβ_{1–42} was reconstituted in DMSO to a stock volume of 250 μM. The peptides were further prepared by sonicating for 10 min followed by centrifuging at 13,362*g* (12,000 rpm) for 15 min. A concentration of 0.5 μM Aβ_{42} was used to treat primary neurons.

## Western blot

Western blot analysis was performed on astrocyte-conditioned media collected from ApoE KO, ApoE3-KI, and ApoE4-KI primary astrocytes to confirm the presence of human ApoE in ApoE3 and ApoE4 astrocyte-conditioned media. The medium was directly collected from astrocytes and centrifuged at 10,000*g* for 10 min at

**Antibodies and reagents.**

| Antibodies | Company | Identifier |
|---|---|---|
| DAPI (4′,6-diamidino-2phenylindole) | Sigma-Aldrich | Cat#D9542 |
| Mouse monoclonal anti-APP/Aβ (6E10) | BioLegend | Cat#SIG-39320; RRID: AB_662798 |
| Mouse monoclonal anti-N-terminal Aβ (82e1) | IBL international | Cat#10323; RRID: AB_10707424 |
| Mouse monoclonal anti-Aβ 1-42 (12F4) | Covance | Cat# SIG-39142 |
| Rabbit polyclonal anti-amyloid fibrils OC | Millipore | Cat#AB2286; RRID: AB_1977024 |
| Rabbit monoclonal anti-ApoE (16H22L18) | Thermo Fisher Scientific | Cat#701241; RRID: AB_2532438 |
| Goat polyclonal anti-ApoE | Thermo Fisher Scientific | Cat#PA1-26902; RRID: AB_779281 |
| Mouse monoclonal anti-β-actin | Sigma-Aldrich | Cat# A5316; RRID: AB_476743 |
| Rabbit polyclonal anti-early endosomal antigen 1 (EEA1) | Sigma-Aldrich | Cat# E4156; RRID: AB_609870 |
| Mouse monoclonal anti-GM130 | BD Biosciences | Cat#610822; RRID: AB_398141 |
| F-Actin (Rhodamine phalloidin) | Life Technologies | Cat#R415 |
| Filipin | Sigma-Aldrich | Cat# SAE0088 |
| Mouse monoclonal anti-glial fibrillary acidic protein (GFAP) | Sigma-Aldrich | Cat# G3893; RRID: AB_477010 |
| Mouse monoclonal anti-Iba1 | Millipore | Cat# MABN92; RRID: AB_10917271 |
| Rabbit polyclonal anti-LAMP1 | Abcam | Cat#ab24170; RRID: AB_775978 |
| Rat monoclonal anti-LAMP1 | Abcam | Cat# ab25245; RRID: AB_449893 |
| Rabbit polyclonal anti-LC3β | Cell signaling | Cat#2775; RRID: AB_915950 |
| Chicken polyclonal anti-MAP2 | Abcam | Cat# ab92434; RRID: AB_2138147 |
| Mouse monoclonal anti-neurofilament (SMI312) | Covance | Cat#SMI-312R; RRID: AB_2314906 |
| Rat monoclonal anti-CD140a | BD Biosciences | Cat#558774; RRID: AB_397117 |
| Mouse monoclonal anti-Rab7 | Abcam | Cat# ab50533; RRID: AB_882241 |
| Rabbit monoclonal anti-S100β | Abcam | Cat#ab52642; RRID: AB_882426 |
| Mouse monoclonal anti-TGN38 | Santa Cruz Biotechnology | Cat#Sc-166594; RRID: AB_2287347 |
| Polyclonal DyLight 405-AffiniPure Goat Anti-Chicken IgY (IgG) | Jackson ImmunoResearch | Cat#103-475-155; RRID: AB_2337389 |
| Polyclonal Alexa Fluor 488-AffiniPure Goat Anti-Rabbit IgG | Jackson ImmunoResearch | Cat#111-545-144; RRID: AB_2338052 |
| Polyclonal Alexa Fluor 568 Conjugated Goat Anti-Mouse IgG | Molecular Probes | Cat#A-11004; RRID: AB_2534072 |
| Polyclonal Alexa Fluor 568 Conjugated Goat Anti-Rat | Molecular Probes | Cat#A-11077; RRID: AB_141874 |
| Polyclonal Alexa Fluor 647-AffiniPure Donkey Anti-Chicken IgY (IgG) | Jackson ImmunoResearch | Cat#703-605-155; RRID: AB_2340379 |
| Polyclonal Alexa Fluor 647-AffiniPure Goat Anti-Mouse IgG | Jackson ImmunoResearch | Cat#115-605-003; RRID: AB_2338902 |
| Polyclonal Alexa Fluor 647-AffiniPure Goat Anti-Rat IgG | Jackson ImmunoResearch | Cat#112-605-003; RRID: AB_2338393 |
| Polyclonal HRP-conjugated Goat Anti-Rabbit IgG | R&D Systems | Cat# HAF008; RRID: AB_357235 |
| Polyclonal HRP-conjugated Goat Anti-Mouse IgG | R&D Systems | Cat#HAF007; RRID: AB_357234 |

4°C. The supernatant was collected and directly used or stored at –80°C. Media samples were prepared for loading by mixing with NuPage-reducing agent (NP004; Invitrogen) and Novex NuPage LDS sample buffer (NP007; Invitrogen). Subsequently, the samples were heated at 70°C for 10 min and centrifuged before loading in a NuPAGE 4–12% Bis-Tris gel (NP0321BOX; Invitrogen).

Western blot was also performed on N2a APP$_{Swe}$ cells treated with ApoE astrocyte-conditioned media to determine the APP and Aβ protein levels after human ApoE treatment. 4 h after astrocyte media treatment, media of N2a cells were first collected and N2a cells were washed in PBS, gently scraped, collected in a 1.5 ml tube, and centrifuged at 10,000$g$ for 2 min. The supernatant was removed and the pellet was either snap frozen at –80°C in liquid nitrogen until further use or directly used for Western blot. To detect Aβ using Western blot, lysate samples were lysed in PBS containing 6% SDS and 1% β-mercapto-ethanol, followed by sonication, heating at 95°C for 6 min, and centrifugation at 13,362$g$ (12,000 rpm) for 10 min. To detect Aβ, the N2a lysate samples were mixed with Novex Tricine SDS Sample Buffer (LC1676; Invitrogen), boiled at 95°C for 5 min, shortly centrifuged, and loaded onto a Novex 10–20% Tricine gel (EC6625BOX).

After SDS–PAGE gel electrophoresis, the proteins were transferred to a PDVF membrane using iBlot 2. For APP and Aβ detection, the membranes were subsequently washed in PBS and cut into two (cut was made around 16 kD). The membrane part containing <16 kD proteins was boiled for 5 min in PBS to improve the detection of Aβ. The membrane part containing >16 kD proteins was stained for total proteins (AC2225; Azure) according to the manufacturer's instructions and images were acquired using a Sapphire Biomolecular imager (Azure Biosystems). After total protein quantification, the top part of the membrane (>16 kD) was further cut just below 78 kD to obtain a total of three membrane parts (Fig S6E and F). All membranes were blocked in PBS-T containing 5% skim milk powder. The membranes were incubated overnight with primary antibodies at 4°C, followed by secondary HRP-conjugated antibodies for 1 h at room temperature. All washes were done in PBS-T. The membranes were developed using ECL Substrate and visualized using a Sapphire Biomolecular imager.

Quantification of the bands was performed using Image Lab 6.1. All bands were normalized to total protein and β-actin. For protein quantification in media, the bands were corrected for both total proteins in media and β-actin and total protein from its corresponding lysate to correct for both protein concentrations in the medium and differences in cell number. All bands are presented normalized to the control, where the control condition is untreated N2A APP$_{Swe}$ cells that were used to establish a baseline for APP metabolism.

### Mesoscale analysis

Secreted Aβ$_{40}$ and Aβ$_{42}$ levels were measured using Meso Scale Discovery. Mouse primary brain cultures from WT or APP/PS1 incubated for 24 h with control (fresh neurobasal media), ApoE3 KI or ApoE4 KI astrocyte-conditioned media were collected. Aβ levels in media were assessed with the 4G8 kit (#K15199E) following the manufacturer's protocol. Plates were read using a QuickPlex Q120 reader (Meso Scale Diagnostics). Total protein in cell lysates was determined by BCA assay and used to normalize the Aβ levels to the concentration of total proteins.

### Immunofluorescence

N2a cells and primary neurons (19 DIV) grown on glass coverslips were fixed in 4% PFA, 4% sucrose in PBS for 15 min at room temperature. The cells were incubated in PBS containing 1% BSA, 0.1% saponin (84510; Sigma-Aldrich), and 2% normal goat serum (005-000-121; Jackson ImmunoResearch) for 1 h at room temperature to permeabilize the cells and block unspecific signals. Afterwards, the coverslips containing N2a cells or primary neurons were incubated with primary antibodies, diluted in 2% normal goat serum in PBS, overnight at 4°C, followed by incubation with secondary fluorescently conjugated antibodies for 1 h at room temperature in the dark. The coverslips were mounted on glass slides with ProLong Diamond Antifade Mountant (P36961; Invitrogen). All washes were performed in PBS.

### Epifluorescence microscopy

Microscopy was performed on coverslips containing fluorescently labeled N2a cells or primary neurons. In our experiments, two different epifluorescence microscopes were used: 1. Olympus IX70 microscope equipped with a 405, 488, 568, and 647 nm channel, X-Cite 120Q excitation light source (Excelitas Technologies), a C11440 ORCA-Flash4-oIT digital camera, and 40x NA 1.3 and 60x NA 1.4 oil immersion objectives. An additional 1.5 magnification of the 60x objective was used when taking the images. 2. Nikon Eclipse 80i upright microscope (RRID:SCR_015572) equipped with a 10x (Nikon plan apo, NA 0.45), 20x (Nikon plan apo, NA 0.75), 40x (Nikon plan apo, NA 1.0, oil immersion), and 60x objectives (Nikon apo VC, NA 1.40, oil immersion). The Nikon microscope was connected to a computer containing Nikon Instruments Software-Elements Advance Research (NIS-Elements AR) version 3.2. The focus of the images was set based on the signal in the ApoE channel.

### Confocal microscopy

To obtain image stacks a laser scanning confocal microscope Leica TCS SP8 (RRID: SCR_018169; Leica Microsystems) in combination with Leica Application Suite X software version 3.4.7.23225 (RRID: SCR_013673; Leica Microsystems) was used. A Z-step size of 0.5 μm was used for imaging stacks. Orthogonal images were obtained by using Bitplane Imaris viewer version 9.5.1 (RRID: SCR_007370; Oxford Instruments).

### Image analysis and quantification

To analyze the added ApoE in N2a cells, the ApoE signal was normalized to control (vehicle control or ApoE KO-treated N2a cells). The percentage of N2a cells that take up ApoE was calculated by separating ApoE-positive (at least one ApoE puncta) from ApoE-negative cells (no ApoE puncta). Three criteria were set to determine which N2A cells in each image were analyzed. First, the cells should show ApoE uptake. Second, the cell outliner could be distinguished by the cellular marker phalloidin, and third, the whole cell body should be present in the image. All further analyses on added ApoE were performed on ApoE-positive cells. The number of ApoE puncta and the ApoE puncta area per N2a cell were analyzed using particle analyzer in Fiji ImageJ.

The co-localization of added recombinant ApoE3 and ApoE4 with sub-cellular markers Rab7, LAMP1, GM130, and TGN38 in N2a cells (Figs 1 and S2) was analyzed by determining the percentage of ApoE pixels that co-localized with each marker using Fiji ImageJ. The images were preprocessed using background subtraction and brightness processing to remove unspecific signals. For Rab7, which was detected as small puncta with high background in the cell, the images were also sharpened before background subtraction. Preprocessing settings of ApoE were set based on control-treated N2a cells to avoid endogenous mouse ApoE. After preprocessing of ApoE and the subcellular markers in the images, the images were thresholded and a selection was created for ApoE. Subsequently, the pixel co-localization was measured in Fiji ImageJ. N2a cells containing 5–20 ApoE puncta per cell were included in the pixel co-localization analyses.

To study whether ApoE in neurons is degraded by lysosomes, lysosomal activity was inhibited using Bafilomycin A1 (Fig 2K–M). Images taken from bafilomycin- and ApoE-treated neurons were analyzed using Fiji ImageJ. First, 10 different, nonoverlapping regions of interest were selected along MAP2-positive neurites for each analyzed neuron. The area of each individual region of interest was kept constant. Regions of interest of neurites were selected in such a way that all intracellular ApoE will be detected, meaning that synaptic ApoE puncta proximal to but not overlapping with MAP2 labeling was also included in the analysis. Within these regions of interest, the co-localization between ApoE and LAMP1 was analyzed in the same way as described for N2a cells (previous section). ApoE puncta size was analyzed using particle analyzer.

The number of MAP2-positive neurons and GFAP-positive astrocytes in our primary cultures (Fig 3C) was determined by manual counting using the Cell Counter plugin in ImageJ (https://imagej.nih.gov/ij/plugins/cell-counter.html). DAPI nuclei were used to determine the total number of cells in the culture.

The antibody 82e1 puncta number and size in APP/PS1 primary neurons were analyzed using particle analyzer in imageJ. Regions of interest of the same size were set for 10 neurites in each neuron (based on MAP2 labeling). Background subtraction and brightness adjustment were done for both the 82e1 and ApoE signals. The 82e1 signal was corrected for the signal in WT neurons, ApoE signal was corrected for the signal in ApoE KO treated neurons. 82e1 signal was thresholded and a selection was created. 82e1-ApoE co-localization was measured through pixel co-localization of the percentage of the 82e1 selection that contained ApoE-positive pixels.

Intraneuronal A$\beta_{42}$ levels (Fig 6C and E) were calculated based on the intensity of A$\beta_{42}$-specific antibody 12F4 as previously described by Ubelman et al (10). Regions of interest were set for cell bodies and neurites based on MAP2 labeling using the polygon tool in ICY (https://icy.bioimageanalysis.org/). The mean intensity of 12F4 labeling was measured for the selected cell bodies and neurites and were corrected for the mean intensity of background signal. For one embryo, five independent neurons were analyzed. In each analyzed neuron, the three most prominent neurites were included in the analysis. All mean intensities measured in this study were normalized to the mean intensity measured in APP KO cell bodies and neurites.

The levels of internalized synthetic A$\beta_{42}$ in primary neurons were analyzed based on 6E10 antibody labeling (Fig 7C and D). The 6E10 antibody is known to exclusively label human A$\beta$ and not endogenous mouse A$\beta$. For quantification of the number of 6E10 puncta and the area of the 6E10 puncta, images obtained by epifluorescence microscopy were preprocessed and thresholded using Fiji ImageJ. 10 regions of interest of neurites per cell were selected using the polygon tool in ImageJ based on MAP2 labeling. Both number and size of the puncta were analyzed using particle analyzer. The puncta numbers were corrected for the total area analyzed to have all values corrected to a set area of 1,000 $\mu m^2$.

### Statistical analyses

All statistical analyses were performed in Graphpad Prism 8.4.1. Before statistical testing, it was assessed whether the data were normally (Gaussian) distributed based on normality tests: Shapiro–Wilk tests and Kolmogorov–Smirnov tests, and QQ plots. In case of a normal distribution of the data, unpaired $t$ test was performed to compare ApoE3 with ApoE4 (Figs 1C, I, and J and 4I and J) and one-way ANOVA to compare more than two groups (Figs 5C–E, I, and J, 7C and D, and S7B and C). When the data were not normally distributed, nonparametric Mann–Whitney tests were performed to test statistical difference between ApoE3 and ApoE4 (Figs 2K and L, 4H, and S2B and D). When more than two groups were compared, Kruskal–Wallis tests were used to analyze non-normalized data (Figs 2M, 5F, H, and K, 6C and E, S6C and D, and S7A). All data displayed in graphs were shown as mean ± SD. Big data points in graphs reflect the number of embryos (N), smaller data points reflect the single neurons/cells analyzed (n). In the case of neurites, 10 neurites per neuron were analyzed unless stated otherwise.

## Supplementary Information

## Acknowledgements

We thank Bodil Israelsson, Lund University, for her support with the animal experiments and genotyping of mice. We also thank MultiPark for the use of the confocal microscopy and Imaris software facilities. This project was funded by the European Union Horizon 2020 Research and Innovation Program SYNDEGEN (Marie Skłodowska-Curie grant agreement No. 721802), Innovation Fund Denmark (BrainStem; 4108–00008 A), the Swedish Research Council grant (No. 2019-01125), Alzheimerfonden, Hjärnfonden, and Konung Gustaf V:s & Drottning Victorias Stiftelse.

### Author Contributions

SC Konings: conceptualization, data curation, formal analysis, supervision, validation, investigation, visualization, methodology, project administration, and writing—original draft, review, and editing.
E Nyberg: formal analysis, investigation, visualization, and writing—review and editing.
I Martinsson: investigation and writing—review and editing.
L Torres-Garcia: investigation and writing—review and editing.
O Klementieva: resources, funding acquisition, and writing—review and editing.
C Guimas Almeida: conceptualization, resources, supervision, funding acquisition, project administration, and writing—review and editing.
GK Gouras: conceptualization, resources, supervision, funding acquisition, project administration, and writing—review and editing.

### Conflict of Interest Statement

The authors declare that they have no conflict of interest.

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
