## [Reviewer comments · Life Science Alliance]

Life Science Alliance

Apolipoprotein E intersects with amyloid- β within neurons

Sabine Konings, Emma Nyberg, Isak Martinsson, Laura Torres-Garcia, Oxana Klementieva, Claudia Guimas Almeida, and Gunnar Gouras

DOI: <https://doi.org/10.26508/lsa.202201887>

Corresponding author(s): Gunnar Gouras, Lund University

Review Timeline:

Submission Date:	2022-12-23
Editorial Decision:	2023-02-06
Revision Received:	2023-05-06
Editorial Decision:	2023-05-24
Revision Received:	2023-05-30
Accepted:	2023-05-31

Transaction Report:

February 6, 2023

Re: Life Science Alliance manuscript #LSA-2022-01887-T

Prof. Gunnar K. Gouras
Lund University
Experimental Medical Science
Experimental Dementia Research Unit
Sölvegatan 19, BMC B11
Lund 22184
Sweden

Dear Dr. Gouras,

Thank you for submitting your manuscript entitled "Apolipoprotein E intersects with amyloid- β within neurons" to Life Science Alliance. The manuscript was assessed by expert reviewers, whose comments are appended to this letter. We invite you to submit a revised manuscript addressing the Reviewer comments.

Thank you for this interesting contribution to Life Science Alliance. We are looking forward to receiving your revised manuscript.

Sincerely,

Eric Sawey, PhD
Executive Editor
Life Science Alliance
<http://www.lsa-journal.org>

B. MANUSCRIPT ORGANIZATION AND FORMATTING:

Reviewer #1 (Comments to the Authors (Required)):

In their manuscript Konings and colleagues examined intracellular co-localization of A β and apoE in several types of cells internalizing and/or metabolizing either molecule. They determined astrocytic ApoE localizes mostly to lysosomes in neuroblastoma cells and astrocytes, while in neurons it preferentially localizes to endosomes and autophagosomes of neurites. They explained these differences by the fact that neurons are highly polarized cells. Furthermore, they showed In AD transgenic neurons expressing APP, astrocyte-derived ApoE intersect intracellularly with amyloid precursor protein and with A β . The authors also determined ApoE4 increases the levels of endogenous and internalized A β 42 in neurons. Overall, this is an important study performed using state of the art cell culture and cellular imaging techniques. It clarifies several intracellular mechanisms pertinent to apoE and A β metabolism and points out key differences between various in vitro cellular models widely used to study biology of apoE, A β , and APP. There are several relatively minor comments, which should be addressed prior to the publication.

1. The result section is missing any references to quantitative data, which were obtained as a part of this work and included in multiple figures.
2. Figure captions concerning quantitative analysis are missing any information about number of data point analyzed/displayed and what statistical test was run to analyzed differences across the groups.
3. Some figures contain schematic overview of the experiment, which is a very friendly visual guide for the reader, but some of the figures do not. Maintaining consistence throughout the paper would be a good idea.
4. Since the authors used confocal imaging as the primary research tool and did not characterize biochemical changes imposed on A β by apoE, they used the word apoE/A β intersection rather than interaction, which is acceptable. However, there are previous studies e.g. already cited Kuszczuk et al AJP 2012, which looked into the effects of apoE on biochemical alterations into intraneuronal A β characteristics and provided evidence for direct apoE/A β interaction inside the neurons. It would be useful and beneficial to widen the Discussion by commenting on these previous findings in greater depth especially they are complimentary to the authors findings.

Reviewer #2 (Comments to the Authors (Required)):

In the current manuscript by the Gouras group, the authors explored localization of internalized human ApoE within 3 systems: AD transgenic neurons, astrocytes and neuroblastoma cells. Results show differential localization in the three systems, with co-localization with APP/A β only in neurons, and localized to the lysosome and neurites. APOE genotype effects on intracellular A β were seen when neurons, not astrocytes, differed in their expression of ApoE isoforms. The paper is a welcome addition to the literature because intracellular trafficking of ApoE and A β are not well-studied. And the differences between the model systems are important observations, but enthusiasm is limited by the following questions.

- 1) What is the percentage of cells/dendrites with ApoE puncta (ApoE3 vs ApoE4)? What is the average puncta number per cell/dendrite and puncta size by ApoE genotype (confocal)?
- 2) It's not described clearly how many independent experiments were performed, how many random areas per coverslip were selected and how many ApoE-positive cells/dendrites were analyzed in those areas. The following phrase - "All data displayed in graphs were shown as mean {plus minus} SD. Big data points in graphs reflect number of embryos (N), smaller data points reflect single neurites/cells analyzed (n)" is not clear and does not seem to explain N2A experiments. Please clarify the analysis details in the Method detail section and add corresponding information (number of independent experiments and number of cells/dendrites analyzed per experiment) to the figure legends.
- 3) It looks like only 5-8 cells/neurites were analyzed per each "big data point", which is not enough to draw a conclusion. If my understanding of the number of data points is correct then additional data points (at least 20 of ApoE positive cells/dendrites randomly imaged across the coverslip per an independent experiment) need to be analyzed and the authors should consider using confocal images which are more suitable for colocalization analysis than widefield/epifluorescence microscopy.
- 4) When looking at at Fig.1G (Rab7) and Suppl. Fig 2B (GM130) side-by-side (below), it is hard to explain why 15% ApoE puncta

is considered to be colocalized with Rab7 but no clear colocalization was reported between ApoE and GM130, some of the cells even show higher % of colocalization with GM130 comparing to Rab7.

5) What kind of primary antibodies were used for ApoE WB analysis (Fig. 1J)? If it was the same 16H22L18 antibodies which were used for ICC then the bands contain both mouse and human ApoE, so it is not clear if cells were treated with equal amounts of ApoE3 and ApoE4 in the downstream experiments. Moreover, ApoE4 overexpression can have a toxic effect on cells, if viability assay was not performed it is hard to tell if the media was collected from the same number of cells. Thus, it would be useful to evaluate concentrations of human ApoE in ApoE3 and ApoE4 media, using for example ApoE human ELISA kit (ThermoFisher) or some other approaches.

6) No quantification is presented for ApoE/LAMP1 and ApoE/Rab7 colocalization for N2A cells treated with the cell culture media from KI astrocytes. At a glance there is more colocalization between ApoE3 and Rab7 than ApoE4 and Rab7, moreover, ApoE puncta look larger in the ApoE4-treated group but quantification should be presented.

7) Increasing sample size may lead to significant differences between ApoE3 and ApoE4 as well as ApoE3 and ApoE3/Bafilomycin A groups (Fig. 2 J and K). Also, ApoE4 puncta look notably larger than ApoE3 in some images and might be worth analyzing. Interestingly, a lot of puncta are located along the MAP-2 positive dendritic shafts but not co-localized with it, resembling localization in dendritic spines. Dendritic spines are highly enriched with F-actin and can be easily stained with Phalloidin and co-localization can be evaluated using confocal microscopy.

8) Quantitative analysis of Abeta (APP, APP-betaCTF)/ApoE and ApoE double-positive puncta in ApoE3 and ApoE4 groups should be provided as well as the percentages of co-localization.

9) Might the presence of mouse ApoE in astrocytic media mask some differences in ApoE3 and ApoE4 effects? Overall targeted replacement mice may represent a better model compared to KI mice.

10) The difference between ApoE KO and ApoE4 KI neurons appears to show an effect that may be statistically significant given the average + standard deviation differences between the group. (Especially given ApoE3 and ApoE4 KI show a statistical difference and their average difference+standard deviation appears larger.)

11) The text of the paper reads that Figure 5I is measuring sAPP α , while the figure itself mentions that APP generally is being measured. If the sAPP α band using western blotting was specifically measured in 5I, the graph maybe should reflect this?

12) Text of Supp Figure 5B states the scale bar is 15um, while the images show the scale bar is 10um.

13) Top of Page 12: "Supplementary Figure 9A-D" should be "Supplementary Figure 6A-D".

Reviewer #3 (Comments to the Authors (Required)):

ApoE4 influences various potentially pathogenic processes including accelerating multiple aspects of amyloidosis but there is little understanding of its possible influences on the earliest intracellular stages of amyloidosis, which have recently been recognized to be the critical initiating stage of the process. The authors use various models to investigate the interactions of lipidated astrocytic APOE internalized into neurons with APP amyloidogenic metabolites and they make a number of important novel observations, including one showing APOE colocalizes with A β /BCTF in specific subcellular compartments after internalization. The colocalization and the possibility that they interact pathologically uncovers new possibilities to explain the pathogenic effect of APOE4 at the earliest stage of amyloid formation within neurons, underscoring the increasingly recognized importance of intracellular mechanisms driving AD pathogenicity, including amyloidosis. Interestingly, they show that primary neurons differ from astrocytes and N2a neuroblastoma cells with respect to which endolysosomal organelles accumulate A β /BCTF IR. Furthermore ApoE4 affects A β 42 internalization, subsequent aggregation and/or degradation in neurons in an allele -selective manner when exogenous A β 42 was delivered although at endogenous levels of APP/A β , the accumulation of A β was APOE allele independent. These and other observations are interesting and novel, but there are significant concerns about some conclusions and their interpretations which need to be addressed.

1. The identity of organelles as endosomes or lysosomes is insufficiently established. There are now multiple recent publications that show that LAMP1 is not a specific lysosome marker and, in fact, labels all of the endolysosomal compartments. Although rab7 is helpful in distinguishing late endosomes, it will not distinguish amphisomes from LE. As importantly, distinguishing lysosomes from autolysosomes using LAMP1 is insufficient and this distinction is significant since recent literature (eg. Lee, Nat. Neurosci, 2022) shows that abeta accumulates mainly in autolysosomes. These are hard to distinguish from other degradative compartments without multiple organelle markers, ideally by triple immunolabeling.

2. Related to the above, the potential role of autophagy dysregulation or deficit as a potential factor in explaining both the cellular localization of Abeta or APOE accumulation or the basis of APOE's effect of promoting abeta accumulation and aggregation is not adequately addressed and recent evidence pertinent to this issue is not cited. Moreover, it has been reported that much of the retrograde "endosome" traffic in axons actually reflects movements of amphisomes which presumably enter the autophagy pathway as they reach the soma. This perhaps is relevant to explaining the differences in organelle localization of APOE or abeta between neurons and astrocytes or N2a in these studies.

3. Discussion is needed about the APOE allele-independence at endogenous levels of A β versus the APOE allele-dependence when exogenous A β 42 is supplied to the cultures. The disease-relevance of this difference should be discussed. Are the levels of exogenous A β 42 applied a proper mimic to levels expected to be in the extracellular space in AD?

4. The observations that the majority of added astrocytic ApoE did not co-localize with either LAMP1- or Rab7-positive puncta, but did colocalize with LC3 is consistent with published data that autophagosomes are formed actively in terminals and either move retrogradely as amphisomes or early autolysosomes. This literature should be discussed. What is more puzzling is the suggestion that this source of ApoE in neurites does not reach lysosomes and the authors' conclusion that lysosomal degradation in neurons is not a major pathway of internalized astrocyte-derived ApoE. This seems unlikely, especially in the

absence of an alternative degradative pathway for vesicular cargoes provided by the authors. A more likely explanation is that the neuritic APOE cargo enters the autophagy pathway and its autolysosomal/lysosomal degradation is delayed beyond the 4h window that the author's used, (which might also explain the lack of effect of Baf). The foregoing is purely a speculative idea to consider. In any case, the authors are encouraged to explore later timepoints to determine the fate of LC3 compartments in the neurites carrying APOE since the implication of a non-lysosomal mechanism for their fate, without experimental support, is not persuasive or biologically straightforward. Equally puzzling is the possibility implied by the authors that the LC3/APOE positive compartment remains in the neurite, unless there is evidence presented of neuritic accumulation of these vesicles.

We are resubmitting our revised manuscript entitled "Apolipoprotein E intersects with amyloid- β within neurons". We thank the reviewers for their efforts in carefully reviewing our work and below address point by point the comments by the reviewers. Added or changed text is highlighted in yellow in our revised manuscript.

Reviewer 1

We were happy to read that the reviewer wrote: *"Overall, this is an important study performed using state of the art cell culture and cellular imaging techniques. It clarifies several intracellular mechanisms pertinent to apoE and A β metabolism and points out key differences between various in vitro cellular models widely used to study biology of apoE, A β , and APP. There are several relatively minor comments, which should be addressed prior to the publication."*

Point 1: *The result section is missing any references to quantitative data, which were obtained as a part of this work and included in multiple figures.*

Response: We now include the quantitative data in the Results section.

Point 2: *Figure captions concerning quantitative analysis are missing any information about number of data point analyzed/displayed and what statistical test was run to analyzed differences across the groups.*

Response: The Figure legends now include information on number of data points and statistical tests used.

Point 3: *Some figures contain schematic overview of the experiment, which is a very friendly visual guide for the reader, but some of the figures do not. Maintaining consistence throughout the paper would be a good idea.*

Response: We had added these schemas for experiments that we had viewed as more complex, while for more straightforward experiments we had omitted them. However, based on the reviewer's point, we now added schemas to all the figures; specifically, the new schemas are in Figure 2H, Figure 3A and Figure 5A.

Point 4: *Since the authors used confocal imaging as the primary research tool and did not characterize biochemical changes imposed on A β by apoE, they used the word apoE/A β intersection rather than interaction, which is acceptable. However, there are previous studies e.g. already cited Kuszczuk et al AJP 2012, which looked into the effects of apoE on biochemical alterations into intraneuronal A β characteristics and provided evidence for direct apoE/A β interaction inside the neurons. It would be useful and beneficial to widen the Discussion by commenting on these previous findings in greater depth especially they are complimentary to the authors findings.*

Response: We agree with this point and have added text to our Discussion (see the beginning of the 4th paragraph of the revised Discussion) to describe prior evidence of actual ApoE/A β interactions. Further, we carefully re-read the excellent paper by Kuszczuk et al., and now describe and cite more findings from their work, such as their in vivo evidence of elevated intraneuronal A β in crosses of ApoE4 with APP/PS1 mutant transgenic mice (end of 3rd paragraph of revised Introduction).

Reviewer 2

Following a succinct summary of our findings, the reviewer writes: *"The paper is a welcome addition to the literature because intracellular trafficking of ApoE and A β are not well-studied. And the differences between the model systems are important observations, but enthusiasm is limited by the following questions."*

Point 1: *What is the percentage of cells/dendrites with ApoE puncta (ApoE3 vs ApoE4)? What is the average puncta number per cell/dendrite and puncta size by ApoE genotype (confocal)?*

Response: We now include these quantitative data asked for by the reviewer in new Figure 1C-E.

Point 2: *It's not described clearly how many independent experiments were performed, how many random areas per coverslip were selected and how many ApoE-positive cells/dendrites were analyzed in those areas. The following phrase - "All data displayed in graphs were shown as mean {plus minus} SD. Big data points in graphs reflect number of embryos (N), smaller data points reflect single neurites/cells analyzed (n)" is not clear and does not seem to explain N2A experiments. Please clarify the analysis details in the Method detail section and add corresponding information (number of independent experiments and number of cells/dendrites analyzed per experiment) to the figure legends.*

Response: We have now provided the numbers of independent experiments and random areas per coverslip selected in the Figure legends and Methods section of the revised manuscript.

Point 3: *It looks like only 5-8 cells/neurites were analyzed per each "big data point", which is not enough to draw a conclusion. If my understanding of the number of data points is correct then additional data points (at least 20 of ApoE positive cells/dendrites randomly imaged across the coverslip per an independent experiment) need to be analyzed and the authors should consider using confocal images which are more suitable for colocalization analysis than widefield/epifluorescence microscopy.*

Response: For the neuron experiments, 5-8 neurons were analyzed per big data point and for each neuron, 10 independent neurites were selected (and averaged per the neuron); thus, a total of 50-80 neurites per big data point were analyzed. A clearer description of the number of neurons and neurites analyzed for each quantification is now added to the figure legends and Methods section (Image analysis and quantification) in the revised manuscript. We always use confocal for imaging co-localization in brain sections and often for N2a cells but epifluorescence is remarkably good for imaging of our low-density primary neurons, which in contrast to the high background of brain have very low and typically just black background. Epifluorescence also provides the full fluorescence signal of our very thin neurites in culture, which are so thin that confocal actually is less able to image them as well.

Point 4: *When looking at Fig. 1G (Rab7) and Suppl. Fig 2B (GM130) side-by-side (below), it is hard to explain why 15% ApoE puncta is considered to be colocalized with Rab7 but no clear colocalization was reported between ApoE and GM130, some of the cells even show higher % of colocalization with GM130 comparing to Rab7.*

Response: When we quantify co-localization between ApoE and subcellular markers, our approach quantifies the percentage of the ApoE pixels that overlap with subcellular marker pixels. In the case of GM130, ApoE is located near GM130 (and to some extent overlaps as seen in the representative images), but since we are quantifying the pixel co-localization, most ApoE pixels are close but not fully overlapping with GM130. Because of this, only a low percentage of the ApoE pixels are co-localizing with GM130, despite ApoE localizing so near to GM130. We do also note in the Results section that ApoE is particularly near to GM130; specifically, we write

“Despite preferential labeling of internalized ApoE3 and ApoE4 near to GM130-labeled cis-Golgi apparatus...”

Point 5: *What kind of primary antibodies were used for ApoE WB analysis (Fig. 1J)? If it was the same 16H22L18 antibodies which were used for ICC then the bands contain both mouse and human ApoE, so it is not clear if cells were treated with equal amounts of ApoE3 and ApoE4 in the downstream experiments. Moreover, ApoE4 overexpression can have a toxic effect on cells, if viability assay was not performed it is hard to tell if the media was collected from the same number of cells. Thus, it would be useful to evaluate concentrations of human ApoE in ApoE3 and ApoE4 media, using for example ApoE human ELISA kit (ThermoFisher) or some other approaches.*

Response: The human APOE targeted replacement mice that we used are knockin and thus, do not express mouse ApoE or overexpress human ApoE3 or 4; they are under the endogenous mouse ApoE promoter. Thus, the ApoE antibody 16H22L18 only sees the human ApoE in the Western blot shown originally in Figure 1J, but now Figure 1M of our revised manuscript. There are similar levels of hApoE3 and hApoE4 in their respective astrocyte conditioned media as seen in this representative Western blot. We always performed Western blot of astrocyte media that we used to make sure equivalent levels were present in all the astrocyte media that we used. The concentration we used was estimated based on comparisons to synthetic ApoE run on WB and thereby used at comparable levels to physiological levels (see also Supplementary figure 1, Konings et al., 2021). Given this point by the reviewer we also added the following sentence in our revised Results section just after mentioning the human ApoE mice: “Note that these human ApoE targeted replacement mice no longer express mouse ApoE.”

Point 6: *No quantification is presented for ApoE/LAMP1 and ApoE/Rab7 colocalization for N2A cells treated with the cell culture media from KI astrocytes. At a glance there is more colocalization between ApoE3 and Rab7 than ApoE4 and Rab7, moreover, ApoE puncta look larger in the ApoE4-treated group but quantification should be presented.*

Response: As noted in response to point 1, above, we now quantified recombinant ApoE3 versus ApoE4 internalization in N2a cells and did not detect a significant difference in puncta area between ApoE3 and E4. Since we saw a similar pattern of labelling with astrocyte derived ApoE we did not separately quantify this and instead showed representative images. We also quantified astrocytic ApoE in primary neurons and saw no significant difference between ApoE 3 and ApoE4 puncta size (see new Figure 2M).

Point 7: *Increasing sample size may lead to significant differences between ApoE3 and ApoE4 as well as ApoE3 and ApoE3/ Bafilomycin A groups (Fig. 2 J and K). Also, ApoE4 puncta look notably larger than ApoE3 in some images and might be worth analyzing. Interestingly, a lot of puncta are located along the MAP-2 positive dendritic shafts but not co-localized with it, resembling localization in dendritic spines. Dendritic spines are highly enriched with F-actin and can be easily stained with Phalloidin and co-localization can be evaluated using confocal microscopy.*

Response: ApoE puncta size after Baf A1 treatment was now quantified (new Figure 2M), but no significant differences between ApoE3 and ApoE4 were detected. We now also performed longer Baf A1 treatments (8 and 24 h co-treatments with ApoE; see Supplementary Figure 3). However, ApoE was difficult to detect in neurites after these longer time points, but was still noted to co-localize with LAMP1 in MAP2-negative cells, then shown to be astrocytes in the next Figure 3. As also described in response to Reviewer 3, point 4 below, the available literature supports that ApoE is mostly recycled and re-secreted by neurons and is degraded by astrocytes. In terms of ApoE outside of the MAP2 positive dendritic shafts, we agree with the reviewer and comment on this in our Discussion, referring also to our prior Konings et al., 2021 paper which showed ApoE associated with synapses, though we had seen no differences between ApoE3 and E4. We now added new Figure 2N to show such ApoE co-labeling with phalloidin.

Point 8: *Quantitative analysis of Abeta (APP, APP-betaCTF)/ApoE and ApoE double-positive puncta in ApoE3 and ApoE4 groups should be provided as well as the percentages of co-localization.*

Response: We now provide this quantitative analysis; see our new subfigures in Figure 4H-J.

9) *Might the presence of mouse ApoE in astrocytic media mask some differences in ApoE3 and ApoE4 effects? Overall targeted replacement mice may represent a better model compared to KI mice.*

Response: As also noted above in point 5, the human ApoE3 and ApoE4 mice that we used are targeted replacement mice from Jackson Labs that have the human ApoE but no longer express the mouse ApoE. We have now also made this clearer in the text of our revised Results section.

Point 10: *The difference between ApoE KO and ApoE4 KI neurons appears to show an effect that may be statistically significant given the average + standard deviation differences between the group. (Especially given ApoE3 and ApoE4 KI show a statistical difference and their average difference+standard deviation appears larger.)*

Response: We agree with the reviewer and have now also included the p value for the difference between ApoE KO compared to ApoE4 in the bar graph of Figure 7. As the reviewer correctly suspected, there is also a statistically significant difference between ApoE KO and ApoE4. Please see our revised Figure 7C.

Point 11: *The text of the paper reads that Figure 5I is measuring sAPP α , while the figure itself mentions that APP generally is being measured. If the sAPP α band using western blotting was specifically measured in 5I, the graph maybe should reflect this?*

Response: We have now specified sAPP α in Figure 5 where before we had only APP.

Point 12: *Text of Supp Figure 5B states the scale bar is 15um, while the images show the scale bar is 10um.*

Response: Thank you, we corrected this.

Point 13: *Top of Page 12: "Supplementary Figure 9A-D" should be "Supplementary Figure 6A-D".*

Response: Thank you, we also corrected this.

Reviewer 3

After summarizing our findings, the reviewer concludes with: *"These and other observations are interesting and novel, but there are significant concerns about some conclusions and their interpretations which need to be addressed."*

Point 1: *The identity of organelles as endosomes or lysosomes is insufficiently established. There are now multiple recent publications that show that LAMP1 is not a specific lysosome marker and, in fact, labels all of the endolysosomal compartments. Although rab7 is helpful in distinguishing late endosomes, it will not distinguish amphisomes from LE. As importantly, distinguishing lysosomes from autolysosomes using LAMP1 is insufficient and this distinction is significant since recent literature (eg. Lee, Nat. Neurosci, 2022) shows that abeta*

accumulates mainly in autolysosomes. These are hard to distinguish from other degradative compartments without multiple organelle markers, ideally by triple immunolabeling.

Response: We completely agree that LAMP1 is not exclusively a marker of lysosomes and were careful to not indicate this in our manuscript. We had also referenced several papers that addressed this important issue of LAMP1 not just labeling lysosomes. However, we did make the assumption that LAMP1 positive vesicles in neurites of primary neurons are mostly not lysosomes, based both on papers such as one by prominent cell biologists (Parton, Simons & Dotti, 1992; reference 40 in our original submission). We now also cite nomenclature based on brain EM evidence as described in the classic textbook of electron microscopy for brain by Alan Peters, Sanford Palay and Henry DeF. Webster entitled "The Fine Structure of the Nervous System". This book describes that lysosomes in normal brain do not localize to axons or dendrites other than the most proximal part of dendrites. Given the reviewer's comment we however now note that LAMP1 positive organelles that we describe in neurites could represent lysosomes or autolysosomes given that primary neurons in culture are not the same as neurons in the normal brain but have been shown to be more stressed. We did use several other subcellular markers relevant to endosome-autophagy-lysosome system, including EEA1, Rab5, Rab7, LC3 β , Cathepsin D, LAMP1, and employed triple labelling in most of our images, but were limited by our best ApoE antibody being a rabbit monoclonal but even more by the fact that, as the reviewer underscores, it is highly difficult to separate amphisomes from late endosomes or late endosomes from lysosomes, given that markers can label either.

Point 2: Related to the above, the potential role of autophagy dysregulation or deficit as a potential factor in explaining both the cellular localization of Abeta or APOE accumulation or the basis of APOE's effect of promoting abeta accumulation and aggregation is not adequately addressed and recent evidence pertinent to this issue is not cited. Moreover, it has been reported that much of the retrograde "endosome" traffic in axons actually reflects movements of amphisomes which presumably enter the autophagy pathway as they reach the soma. This perhaps is relevant to explaining the differences in organelle localization of APOE or abeta between neurons and astrocytes or N2a in these studies.

Response: We agree with the reviewer that autophagy impairment appears to be of major importance and have now highlight this more in our revised Discussion. Furthermore, we cited 2 recent papers from the Nixon lab on autophagy impairment in AD.

Pont 3: Discussion is needed about the APOE allele-independence at endogenous levels of A β versus the APOE allele-dependence when exogenous A β 42 is supplied to the cultures. The disease-relevance of this difference should be discussed. Are the levels of exogenous A β 42 applied a proper mimic to levels expected to be in the extracellular space in AD?

Response: This is an interesting point, although we can only speculate about differences we see with ApoE genotypes in added A β 42 compared to endogenous A β 42. Physiological levels of A β 42 in the brain interstitial fluid are normally significantly lower than the 0.5 μ M which we used when adding A β 42. However, with Alzheimer's it is less clear what the levels of A β 42 are that could potentially be taken up by neurons. Levels of A β 42 in AD brain are elevated about 10,000 times compared to normal brain (as we described in Roos TT et al., Acta Neuropathol, 2021) and locally should be particularly high near plaques. Unlike most in the field who for decades added high levels of A β 42 when modeling AD, we for many years had focused on comparisons between primary wild-type and endogenously elevated A β in

AD transgenic neurons. However, we did realize (and reported in e.g. Willen K et al., 2017) that adding elevated levels of exogenous A β 42 puts it (following internalization) in the same endosome-autophagy-lysosome compartments in which A β 42 has been shown by several groups, including ours and Lee et al., 2022, to accumulate in brains with AD-like β -amyloidosis. It is also possible that because individuals with ApoE4 are fine when young, that it is only with age and elevated A β 42 that the intersection of ApoE4 and A β 42 promotes cellular endosome-autophagy-lysosome dysfunction and thus, elevation of A β 42 in neurons. Based on this point by the reviewer, we now added additional text discussing this issue to paragraph 5 of our revised Discussion.

Point 4: The observations that the majority of added astrocytic ApoE did not co-localize with either LAMP1- or Rab7-positive puncta, but did colocalize with LC3 is consistent with published data that autophagosomes are formed actively in terminals and either move retrogradely as amphisomes or early autolysosomes. This literature should be discussed. What is more puzzling is the suggestion that this source of ApoE in neurites does not reach lysosomes and the authors' conclusion that lysosomal degradation in neurons is not a major pathway of internalized astrocyte-derived ApoE. This seems unlikely, especially in the absence of an alternative degradative pathway for vesicular cargoes provided by the authors. A more likely explanation is that the neuritic APOE cargo enters the autophagy pathway and its autolysosomal/lysosomal degradation is delayed beyond the 4h window that the author's used, (which might also explain the lack of effect of Baf). The foregoing is purely a speculative idea to consider. In any case, the authors are encouraged to explore later timepoints to determine the fate of LC3 compartments in the neurites carrying APOE since the implication of a non-lysosomal mechanism for their fate, without experimental support, is not persuasive or biologically straightforward. Equally puzzling is the possibility implied by the authors that the LC3/APOE positive compartment remains in the neurite, unless there is evidence presented of neuritic accumulation of these vesicles.

Response: There are a few interesting parts to this last point by the reviewer. In our revised Discussion we address these issues raised by the reviewer. As noted in response to point 2 above, we have now added additional references on the importance of autophagy in AD. We also provide more citations relevant to recycling and re-secretion rather than degradation of ApoE, which has been most thoroughly described for hepatocytes; these cells have been extensively studied in relation to their ApoE/lipid biology. An additional study is now also cited for recycling and re-secretion rather than degradation of ApoE in a neuron cell line (Rellin L et al., 2008). As described also in response to Reviewer 2, point 7, we also carried out additional experiments to address whether ApoE might eventually be seen in lysosomes of neurons beyond the 4 h window with now also 8 h and 24 h Baf A1/ApoE treatments; see new Supplementary figure 3. As we noted above, with these longer time points of Baf A1, ApoE levels become much fainter in neurons but are at high levels colocalized with LAMP1 in astrocytes, which appears consistent with ApoE re-secretion from neurons and degradation by astrocytes.

May 24, 2023

RE: Life Science Alliance Manuscript #LSA-2022-01887-TR

Prof. Gunnar K. Gouras
Lund University
Experimental Medical Science
Experimental Dementia Research Unit
Sölvegatan 19, BMC B11
Lund 22184
Sweden

Dear Dr. Gouras,

Thank you for submitting your revised manuscript entitled "Apolipoprotein E intersects with amyloid- β within neurons". We would be happy to publish your paper in Life Science Alliance pending final revisions necessary to meet our formatting guidelines.

- please upload your supplementary figures as single files
- please use the [10 author names, et al.] format in your references (i.e. limit the author names to the first 10)
- please add the panel E to your figure 3 figure legend; please add the panels L and M to the figure 4 legend

A. FINAL FILES:

B. MANUSCRIPT ORGANIZATION AND FORMATTING:

Sincerely,

Reviewer #1 (Comments to the Authors (Required)):

This is a revised manuscript by Konings and colleagues who examined intracellular co-localization of Ab and apoE in several types of cells internalizing and/or metabolizing either molecule. They determined astrocytic ApoE localizes mostly to lysosomes in neuroblastoma cells and astrocytes, while in neurons it preferentially localizes to endosomes and autophagosomes of neurites. They explained these differences by the fact that neurons are highly polarized cells. Furthermore, they showed In AD transgenic neurons expressing APP, astrocyte-derived ApoE intersect intracellularly with amyloid precursor protein and with A β . The authors also determined ApoE4 increases the levels of endogenous and internalized A β 42 in neurons. Overall, this is an important study performed using state of the art cell culture and cellular imaging techniques. It clarifies several intracellular mechanisms pertinent to apoE and Ab metabolism and points out key differences between various in vitro cellular models widely used to study biology of apoE, Ab, and APP. The initial submission contained several minor deficiencies concerning discussion, and consistency and clarity of presenting data which now have been fully addressed. I found the revised manuscript improved and even somewhat expanded.

Reviewer #2 (Comments to the Authors (Required)):

The authors have fully addressed all concerns.

May 31, 2023

RE: Life Science Alliance Manuscript #LSA-2022-01887-TRR

Prof. Gunnar K. Gouras
Lund University
Experimental Medical Science
Experimental Dementia Research Unit
Sölvegatan 19, BMC B11
Lund 22184
Sweden

Dear Dr. Gouras,

Thank you for submitting your Research Article entitled "Apolipoprotein E intersects with amyloid- β within neurons". It is a pleasure to let you know that your manuscript is now accepted for publication in Life Science Alliance. Congratulations on this interesting work.

DISTRIBUTION OF MATERIALS:

Again, congratulations on a very nice paper. I hope you found the review process to be constructive and are pleased with how the manuscript was handled editorially. We look forward to future exciting submissions from your lab.

Sincerely,
